# Deep structured learning for variant prioritization in Mendelian diseases

**Matt C. Danzi** [1], **Maike F. Dohrn** [1,2], **Sarah Fazal**[1], **Danique Beijer** [1], **Adriana P. Rebelo** [1], **Vivian Cintra**[1] & **Stephan Züchner** [1] ✉

Effective computer-aided or automated variant evaluations for monogenic diseases will expedite clinical diagnostic and research efforts of known and novel disease-causing genes. Here we introduce MAVERICK: a Mendelian Approach to Variant Effect pRedICtion built in Keras. MAVERICK is an ensemble of transformer-based neural networks that can classify a wide range of protein-altering single nucleotide variants (SNVs) and indels and assesses whether a variant would be pathogenic in the context of dominant or recessive inheritance. We demonstrate that MAVERICK outperforms all other major programs that assess pathogenicity in a Mendelian context. In a cohort of 644 previously solved patients with Mendelian diseases, MAVERICK ranks the causative pathogenic variant within the top five variants in over 95% of cases. Seventy-six percent of cases were solved by the top-ranked variant. MAVERICK ranks the causative pathogenic variant in hitherto novel disease genes within the first five candidate variants in 70% of cases. MAVERICK has already facilitated the identification of a novel disease gene causing a degenerative motor neuron disease. These results represent a significant step towards automated identification of causal variants in patients with Mendelian diseases.

An estimated 300 million individuals worldwide suffer from rare inherited diseases[1]. Rare diseases cause an enormous burden for affected patients and families as well as for health care providers. Genetic testing plays an important role in the diagnosis of inherited rare conditions; and with many genetic treatments in development, precise genetic diagnosis is increasingly imperative for patient survival and quality of life[2–5]. Thanks to declining sequencing costs, large gene panels, exomes, and whole genomes are now fully implemented into routine clinical genetic work-up strategies. However, the process of classifying DNA variants into a spectrum of benign to pathogenic remains cumbersome and afflicted by uncertainties. Nearly all approaches rely on heuristic considerations of allele frequencies, prior evidence, conservation, inheritance trait, and more. Statistical models have guided development of frameworks for rare variant interpretation[6,7], but ultimately will be limited by the fact that the majority of rare disease-causing variation is only observed once.

Large population-based studies, such as gnomAD or the UK biobank, have revealed that 90% of all genomic variation is uniquely observed or very rare (MAF < 2.4e−4)[8]. These many changes begin to obscure the variants with strong genetic effects. This further necessitates large-scale classification approaches of DNA variants for genotype–phenotype correlations.

Currently, a significant number of patients with rare diseases remain genetically undiagnosed, exposing a large diagnostic gap. A recent study on UK biobank data reported a diagnostic success rate of 16% on 7065 rare disease patients[9]. This situation introduces ambiguity into the diagnostic process and ultimately requires more research into the allelic causes of rare (and common) diseases.

The discovery of new disease-causing genes is generally improved by ever larger genome-scale datasets. Prioritizing among a great number of potential candidate variants remains a major obstacle. In silico variant pathogenicity prediction tools, such as CADD,

[1]Dr. John T. Macdonald Foundation Department of Human Genetics and John P. Hussman Institute for Human Genomics, University of Miami Miller School of Medicine, Miami, FL, USA. [2]Department of Neurology, Medical Faculty of the RWTH Aachen University, Aachen, Germany. ✉e-mail: szuchner@med.miami.edu

MutationAssessor, MutationTaster, SIFT, PolyPhen2, PROVEAN, VEST, and PrimateAI are important[10-18], but they are not sufficient for classifying a single causal variant in a patient with a rare monogenic disease. Many of these programs only assess missense variation, but not frameshift or in-frame indels. Many of the widely used pathogenicity prediction tools are based on statistical learning approaches, but a subset including DANN and PrimateAI have employed deep learning[16,19]. Few Mendelian variant pathogenicity prediction tools have been developed to address this gap, such as MAPPIN and ALoFT, but they each also have a relatively narrow focus[20,21]. Both tools are based on the statistical random forest machine learning approach. Exomiser is a notable tool, designed to identify causal variants in patients with rare monogenic diseases[22,23]. In a prioritization approach, Exomiser uses the non-Mendelian pathogenicity predictors PolyPhen2, MutationTaster, SIFT, and optionally CADD and REMM to score variants. Exomiser additionally incorporates phenotypic HPO terms (Human Phenotype Ontology)[24] in order to score the relevance of genes to the phenotype. It then combines these variant and gene scores to rank all variants in a VCF and stratifies these results by possible modes of inheritance.

In this work, we report the development of a variant pathogenicity prediction software, called MAVERICK. To our knowledge, MAVERICK is the first neural network-based variant pathogenicity prediction tool built specifically for Mendelian disease contexts. The leading classification driver is the DNA variant in its surrounding sequence of 100 amino acids on each side layered with deep annotation information. MAVERICK addresses shortcomings of similar tools by (1) inheritance trait specific pathogenicity−scoring of variants as either pathogenic dominant, pathogenic recessive, or benign; and (2) evaluating a broad set of protein-altering variants that includes missense, nonsense, frameshifting, and non-frameshifting variation. We evaluate MAVERICK's superior performance relative to similar variant pathogenicity prediction tools. We demonstrate that MAVERICK is capable of consistently prioritizing the causal variant among all others in simulated and real patients.

## Results

### Overview of MAVERICK architecture and training

MAVERICK is an ensemble of eight neural networks, which use the transformer as their primary building block. The transformer is a neural network construct that processes sequential input−in this case amino acids of a protein[25]. Figure 1a presents a conceptual overview of MAVERICK's inputs and outputs. The eight ensemble members span two distinct architectures which are described in full detail in the methods section. Briefly, architecture 1 (Supplementary Fig. 1) contrasts the reference with the altered protein sequence and employs evolutionary conservation information as well as structured data on allele frequency, gene constraint, and more[8,26−31]. Architecture 2 (Supplementary Fig. 2) also uses the altered protein sequence, evolutionary conservation data, and structured data, but uses a pre-trained "protein language" model called ProtT5-XL BFD as a second type of feature extractor for the altered protein sequence[32]. Both architectures output a three-class prediction for each variant, which is the probability that a variant is either benign, dominant pathogenic, or recessive pathogenic; such that those three scores always sum up to 1.

Training and testing of MAVERICK paid particular attention to separating the sets of variants used for training from those used for performance evaluation (to prevent circularity). MAVERICK was trained on variants added to ClinVar prior to 2020. Benign variants were drawn from both ClinVar benign variants and variants observed homozygous in gnomAD in more than two individuals. ClinVar pathogenic variants were divided into dominant and recessive groups using their associated OMIM disease terms. Hyperparameter tuning was performed using a validation set of 1000 variants held out from the training set. MAVERICK has primarily been evaluated using two datasets (Fig. 1b): the known genes test set (16,012 novel variants on known disease genes), and the novel genes test set (1930 variants on novel disease genes). The test sets were created in the same way as the training set, just using a more recent release of ClinVar and removing all variants seen in the training and validation sets[33]. Thus, the known genes set contains variants identified as pathogenic or benign in

**a**

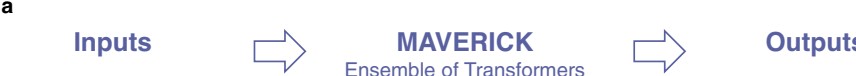

| Inputs | MAVERICK Ensemble of Transformers | Outputs |
|---|---|---|

- Protein sequence 201aa reference vs alteration
- Genetic constraint
- Evolutionary conservation
- Allele frequency, GERP, GDI, RVIS, pext scores

- 2 Architectures
- 8 Models

- Probability variant is dominant
- Probability variant is recessive
- Probability variant is benign

**b**          **Datasets used in the manuscript**

**Training dataset**:
126,739 variants added to Clinvar before 2020

**Known genes testing dataset**:
16,012 variants added to Clinvar during 2020 on genes present in the training set

**Validation dataset**:
1,000 variants added to Clinvar before 2020

**Novel genes testing dataset**:
1,930 variants added to Clinvar during 2020 on genes not present in the training set

**Fig. 1 | Conceptual overview of MAVERICK. a** Diagram of MAVERICK inputs and outputs. MAVERICK takes as inputs the reference and altered protein sequence, the evolutionary conservation of each amino acid in the protein, and structured data including genetic constraint and allele frequency information. These inputs are then processed through MAVERICK's ensemble of transformer-based neural networks to produce the output: a three-class prediction corresponding to the probability that the input variant is benign, pathogenic with dominant inheritance, or pathogenic with recessive inheritance. The three output probabilities always sum to one. **b** MAVERICK training and testing datasets. MAVERICK's training and validation datasets were created from variants in ClinVar prior to the year 2020. The known and novel genes test datasets were created from variants added to ClinVar during 2020, following the same rules for creation as the training set.

ClinVar in the year 2020 on genes that have at least one pathogenic variant in the training set. Similarly, the novel genes set contains variants identified as pathogenic or benign in ClinVar in the year 2020 on genes that did not have any pathogenic variants in the training set. The genes composing the novel genes set may or may not have been scientifically novel in 2020, but they are novel to MAVERICK, given what it saw in its training set. This is meant to simulate the discovery of novel disease genes. These test sets are intended to demonstrate MAVERICK's performance in different scenarios with the novel genes test set being the more difficult. See Methods for more details.

## MAVERICK effectively classifies the pathogenicity of a wide range of protein-altering variants

MAVERICK accurately classified the variants in the known genes and novel genes test sets with areas under the receiver operating characteristic (ROC) curve above 0.9 for each Mendelian trait class (Fig. 2a). These datasets are class imbalanced: the known genes set has relatively few benign variants while the novel genes set is mostly composed of benign variants. Thus, the area under the precision-recall curve (auPRC) is a more useful metric of performance than area under the ROC curve, because it reveals MAVERICK's performance on the minority classes rather than allowing its score to always be dominated by the performance of the largest class. Figure 2b plots the area under the precision-recall curve for each dataset, revealing how much more challenging the novel genes test set is, particularly for identifying dominant variants. MAVERICK achieves over 0.94 in auPRC for each Mendelian trait class in the known genes set. MAVERICK shows its lowest performance of 0.63 in the dominant variants class in the novel genes set, but that auPRC score is still well above the random guessing threshold of 0.1 for those variants. The random guessing threshold is 0.1 for this measure because dominant variants compose ~10% of the novel genes test set. Precision, recall, and F1 scores for each class in each dataset are provided in Supplementary Table 1, along with the values plotted in Fig. 2a, b. The distribution of scores for each class of variants in each test set are shown as violin plots in Supplementary Fig. 3A, B. Taken together, these results indicate that MAVERICK largely succeeds at assessing for Mendelian pathogenicity as a three-class classifier.

We next sought to quantify the contribution of each of MAVERICK's sources of input to its overall performance. We performed a series of ablation experiments, individually dropping all edges in the neural networks that carried each type of input. Supplementary Fig. 3C reveals that MAVERICK is quite robust to the loss of any individual source of information. The structured data (e.g., allele frequency, gene constraint) is the most indispensable, causing up to a 27% decrease in performance when dropped out. The reference sequence and the encoding of the altered sequence with ProtT5-XL BFD cause little change to the performance of the ensemble when they are ablated. The ablation of all sources of input is also plotted to show the random guessing performance for each dataset.

The ablation experiments demonstrated MAVERICK's reliance on the structured data, much of which is identical among different variants on the same gene. This makes possible information leakage between the training set and novel variants on genes present in the training set, such as those in the known genes test set. Therefore, it is important to robustly measure MAVERICK's performance on variants from novel disease genes in order to accurately estimate it's abilities independent of any information leakage, which should give insight into its true generalization potential. MAVERICK has already been assessed with the novel genes test set for this purpose, but since there were only 345 genes with pathogenic variants in that test set, it is possible that this small group was non-representative. To provide a more robust estimate of MAVERICK's performance for genes on which it was not trained, we trained an alternative version, termed CV-MAVERICK, which performed the entire training of MAVERICK as a five-fold cross-validation loop where each fold contained the variants from 20% of the

genes in the training set. In this way, all variants in each held-out fold were on genes completely novel to CV-MAVERICK—it had never been trained on any pathogenic or benign variants on any of the held-out genes. CV-MAVERICK is then an ensemble of 40 models: 5 models each trained on 80% of the data for each of the original eight MAVERICK ensemble members. Therefore, cross-validation performance of CV-MAVERICK represents performance on 1930 disease genes novel to it.

On these held-out genes, CV-MAVERICK achieves auPRC scores greater than MAVERICK achieved on the novel genes test set for each of the Mendelian trait classes (Supplementary Fig. 3D). However, CV-MAVERICK performs slightly worse than MAVERICK on the known and novel genes test sets (Supplementary Fig. 3D). Precision, recall, and F1 scores for each class in each dataset are provided in Supplementary Table 2, along with the values plotted in Supplementary Fig. 3D. Overall, CV-MAVERICK has slightly worse performance than MAVERICK. This is unsurprising since each member of the CV-MAVERICK ensemble was trained on only 80% of the data on which MAVERICK was trained. Yet, CV-MAVERICK maintains similar novel gene performance on a very large set of disease genes novel to it through the cross-validation method. Together, the ablation and cross-validation experiments show that MAVERICK's performance benefits from some information shared among variants on known disease genes, thereby giving it stronger performance on known genes, but it also generalizes well to novel genes with only slightly lower performance, as estimated by CV-MAVERICK's performance on the held-out genes as well as by MAVERICK and CV-MAVERICK's performance on the novel genes test set. The performance of CV-MAVERICK on the cross-validation genes also suggests that the novel genes test set provided a reasonably accurate estimate of MAVERICK's performance on the overall landscape of novel disease genes.

Since MAVERICK is capable of classifying a wide variety of variant types, it is important to ensure that it maintains similar performance among the different categories. MAVERICK performs acceptably across all tested variant types (Fig. 2c, d). We observe that MAVERICK's scores are well-calibrated on the known genes test set (Supplementary Fig. 4A, B). Specifically, squared values of the Pearson correlation coefficient between bins of model confidence and accuracy of predictions within each bin were all above 0.95 and slopes of the fitted linear trend lines were all close to 1 (0.93–1.04). However, MAVERICK's predictions were less well-calibrated on the variants in the novel genes test set. While squared values of the Pearson correlation coefficient between bins of model confidence and accuracy of predictions within each bin were still high, ranging from 0.85–0.97. We see that MAVERICK is over-confident for dominant pathogenic variants on these novel genes, as indicated by the slope of its fitted linear trend line being only 0.63, revealing that MAVERICK's confidence does not grow in accordance with its accuracy on these variants (Supplementary Fig. 4A, B).

To comprehensively test MAVERICK's ability to identify pathogenic variants with the correct mode of inheritance in a disease gene, we scored all possible missense variants on the dominant spastic paraplegia gene *SPAST* (Fig. 2e). *SPAST* was selected because it has a relatively large number of known pathogenic variants associated with the Mendelian spastic paraplegia phenotype. Figure 2e plots MAVERICK's predictions for *SPAST* where each dot is a missense variant, which gets plotted three separate times using the dominant, recessive, and benign scores as the y-axes. The x-axis gives the position of the amino acid within the canonical transcript of the protein. MAVERICK correctly expects missense variants on *SPAST* to cause dominant disease even when they are distant from any known disease-causing variant in ClinVar. Specifically, there are 40 missense variants in *SPAST* labeled as pathogenic in ClinVar which were not part of the training set. Of those, MAVERICK identifies 37 as dominant pathogenic, one as recessive pathogenic, and two as benign. MAVERICK also predicts that variants disrupting certain domains of the protein are more likely to be deleterious than other regions (Fig. 2e).

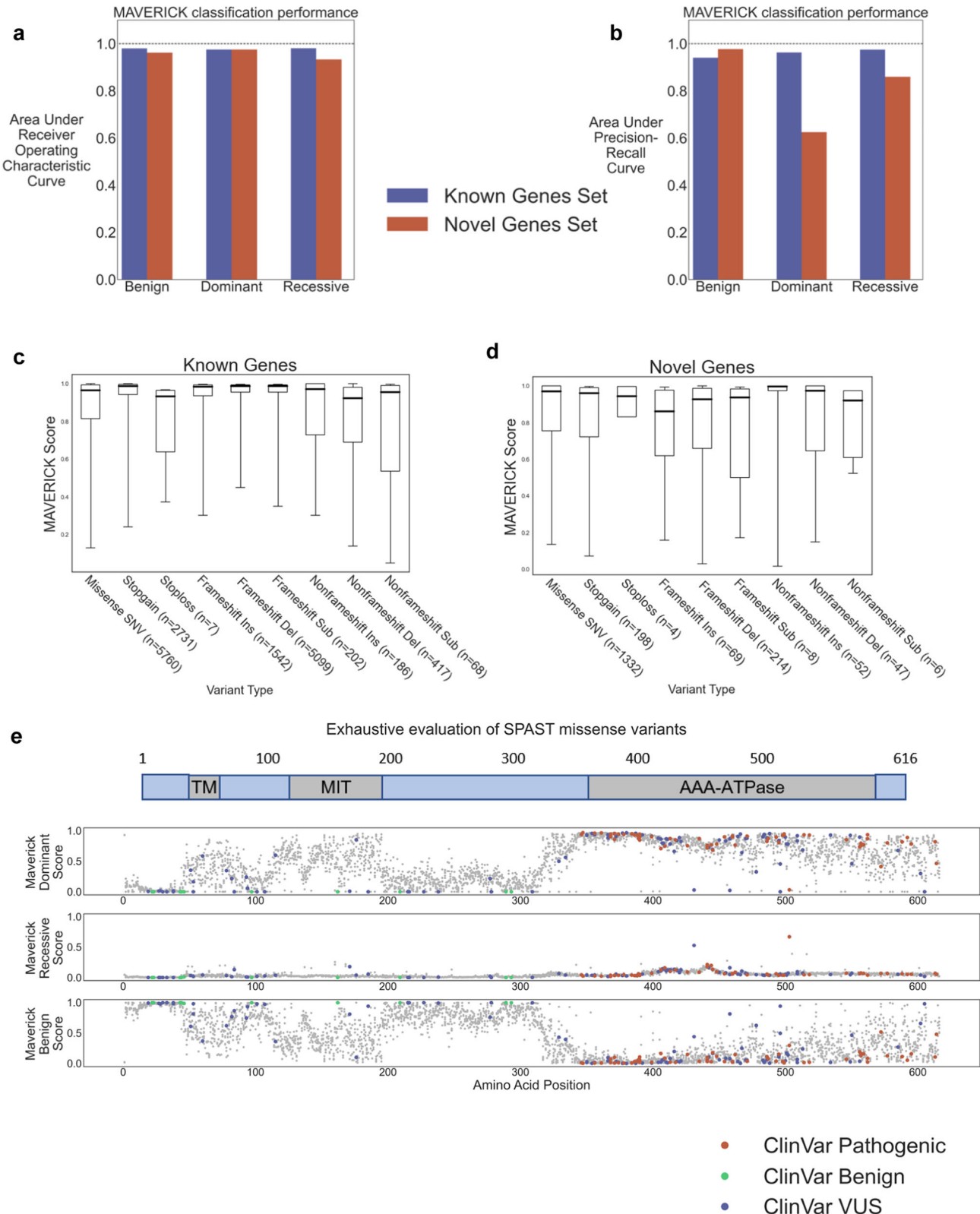

**e** Exhaustive evaluation of SPAST missense variants

Legend:
- ● ClinVar Pathogenic
- ● ClinVar Benign
- ● ClinVar VUS
- · Not in ClinVar

## MAVERICK reliably prioritizes causal variants in simulated cases

The primary purpose for which MAVERICK was developed was to prioritize variants from patients with Mendelian diseases using only genotype information. In this context, there is generally a large excess (often around 400:1) of benign (or recessive variants in the heterozygous state without a compound partner) variants to pathogenic ones. Thus, a useful tool for variant prioritization must have a low false positive rate and predict few pathogenic variants per sample. We tested MAVERICK on 535,292 variants from 10,138 patients with Mendelian diseases from the GENESIS database[34]. MAVERICK predicted a median of six (mean of 7.9) pathogenic variants per sample (Fig. 3a). Under a simplified and conservative model, assuming strict Mendelian

**Fig. 2 | MAVERICK effectively classifies the pathogenicity of a wide-range of protein-altering variants. a** Areas under the receiver operating characteristic curve for the known genes and novel genes test sets. **b** Areas under the precision-recall curve for the known genes and novel genes test sets. **c, d** Box plots of MAVERICK classification performance on each type of protein-altering variant that it can assess. The y-axis shows the distribution of MAVERICK predictions for each variant type where the value plotted for any given variant is the probability for the true class label (e.g., benign variants are plotted by their benign score). The boxes show the median, 25th and 75th percentiles, while the whiskers show the 5th and 95th percentiles and any remaining points are shown as blue dots. The number of variants used for each box plot is shown at the bottom of the figure. **c** The performance for the known genes test set. **d** The performance for the novel genes test set. **e** MAVERICK predictions for every possible missense variant on the known dominant spastic paraplegia gene *SPAST*. For each variant, MAVERICK's predicted benign score is plotted on the bottom subplot, the recessive score is plotted on the middle subplot, the dominant score is plotted on the top subplot. A diagram of domains in the gene is given at the top. ClinVar pathogenic variants are plotted in red. ClinVar benign variants are plotted in green. ClinVar variants of uncertain significance are plotted in blue.

inheritance and only one such disease per person, at most one of these six predictions can be true, so that is five to six false positives per sample. The vast majority of these pathogenic predications are variants that are classified to have dominant effects. Further, 95% of the samples have 15 or fewer predicted pathogenic variants. Combined with the earlier results showing that MAVERICK has reasonable sensitivity for pathogenic variants, these results suggest that it should be useful to prioritize pathogenic variants in patient samples and that the pathogenic coding variant should fall within the top six predictions in at least half of all cases.

To rigorously assess MAVERICK's ability to prioritize causal variants from patients, we first performed spike-in analyses. We collected 98 control samples and tested how MAVERICK would rank each variant in the known and novel genes test sets when added to each control sample relative to all other variants present in that sample: a process we refer to as spiking-in pathogenic variants. Figure 3b shows the results of this spike-in analysis for 13,095 pathogenic variants in the known genes set and 696 pathogenic variants in the novel genes set. This analysis does not use any phenotypic information and includes all inheritance patterns. The results are even better than what was suggested by the false positive analysis: instead of requiring a median of 6 guesses per sample, MAVERICK solves the cases with novel variants on known genes with a median of one guess and it solves the cases on novel genes in a median of three guesses each. In fact, MAVERICK solves 70.1% of the cases on known genes on the first guess and 41.5% of the cases on novel genes on the first guess. By five guesses, MAVERICK solves 89.3% of the cases on known genes and 69.4% of the cases on novel genes. Supplementary Fig. 5A and 5B show the results of the spike-in analysis for the known and novel disease gene sets, respectively, split out between dominant and recessive genes. MAVERICK's performance is similar for dominant and recessive variants, with a small bias toward the recessive variants in these datasets. Supplementary Fig. 5C shows the area under the curve of the cumulative percentage of causal variants identified across the simulated cases within the top 20 variants in each case for each of the individual sub-models that compose MAVERICK's ensemble. The ensemble outperforms each individual sub-model, albeit by a small margin.

Since MAVERICK is a genotype-only pathogenicity classifier, its performance can theoretically be improved considerably by incorporating inheritance and phenotypic information. Figure 3c, d lay out how MAVERICK optionally incorporates inheritance or phenotype information, respectively. Incorporation of inheritance information is a simple filtering step. Incorporation of phenotypic information involves averaging MAVERICK's prediction with a gene-phenotype association score from another tool. Here we use three such tools: GADO, Phenix, and HiPhive[22,35]. Briefly, GADO uses patterns in gene expression data to identify gene-phenotype relationships and is designed to aid novel disease gene discovery; Phenix draws on knowledge of known human Mendelian disease genes; and HiPhive leverages model organism phenotypes and protein-protein interaction data in addition to known disease-gene phenotypes.

As a first assessment of this extended approach, we collected up to five HPO terms associated with the OMIM phenotype of each variant in the known genes and novel genes sets. We then scored every gene

for its relevance to each set of HPO terms using the phenotype prioritization tool GADO[35]. We next repeated the spike-in analyses but averaged each variant's score from MAVERICK with the GADO score for that gene for the set of HPO terms associated with the spiked-in variant. Figure 3e shows that this additional information considerably improved ranking of the variants from both the known genes and novel genes test sets. Over 88% of the known genes set cases and more than half of the novel genes set cases were solved on the first guess when phenotype information was incorporated. Alternatively, with inheritance information, MAVERICK is able to solve 80.4% of the cases on known genes and 68.6% of the cases on novel genes on the first guess. It solves over 90% of the cases on both known and novel genes within 10 guesses. Finally, incorporating both inheritance and phenotypic information led to the strongest performance with MAVERICK able to solve 91.6% of the cases on known genes and 75.4% of the cases on novel genes on the first guess. Within five guesses, MAVERICK solved 96.2% of the cases on known genes and 90.8% of the cases on novel genes.

## MAVERICK outperforms similar tools

To our knowledge, the tools most similar to MAVERICK are MAPPIN and ALoFT. Both of these tools apply a Mendelian approach to evaluating variant pathogenicity. Their main drawbacks are in their limited scope: MAPPIN only scores SNVs, while ALoFT only scores loss-of-function variants. We compared MAVERICK's performance to MAPPIN on the missense, stop-gain and stop-loss SNVs from the known genes and novel genes test sets (Fig. 4a, b, Supplementary Table 3). We found that while MAPPIN and MAVERICK have similar recall for pathogenic dominant and recessive missense SNVs, MAPPIN struggles with benign variants and predicts over 90% of them pathogenic. Overall, MAVERICK's performance for classifying missense SNVs is significantly better than MAPPIN's as measured by area under the precision-recall curve (one-sided Mann–Whitney U Test, *p*-value < 0.0001).

Next, we compared MAVERICK's performance to ALoFT on the loss-of-function variants from the test sets (Fig. 4c, d, Supplementary Table 4) and found that both tools perform similarly. MAVERICK shows a small lead in each comparison of precision, recall, and area under the precision-recall curve for both known and novel genes. MAVERICK's lead in area under the precision-recall curve is statistically significant for each comparison (one-sided Mann–Whitney U Test, *p*-value < 0.0001).

While there are relatively few pathogenicity classification tools that differentiate among pathogenic variants as dominant or recessive for Mendelian contexts, there are many which take the more typical binary classification approach of benign vs pathogenic. These tools may have different assumptions than MAVERICK for non-fully penetrant alleles with small effect sizes which contribute to complex diseases, but they are often used to help identify causal mutations in individuals with rare monogenic diseases similar to how we propose MAVERICK be used. We compared the performance of MAVERICK, MAPPIN, and 36 non-Mendelian pathogenicity classifiers in a version of the variant prioritization task shown in Fig. 3b, e, but using only SNVs in order to maximize comparability across the disparate tools (Fig. 4e, f). Binary versions of MAVERICK and MAPPIN were also included which

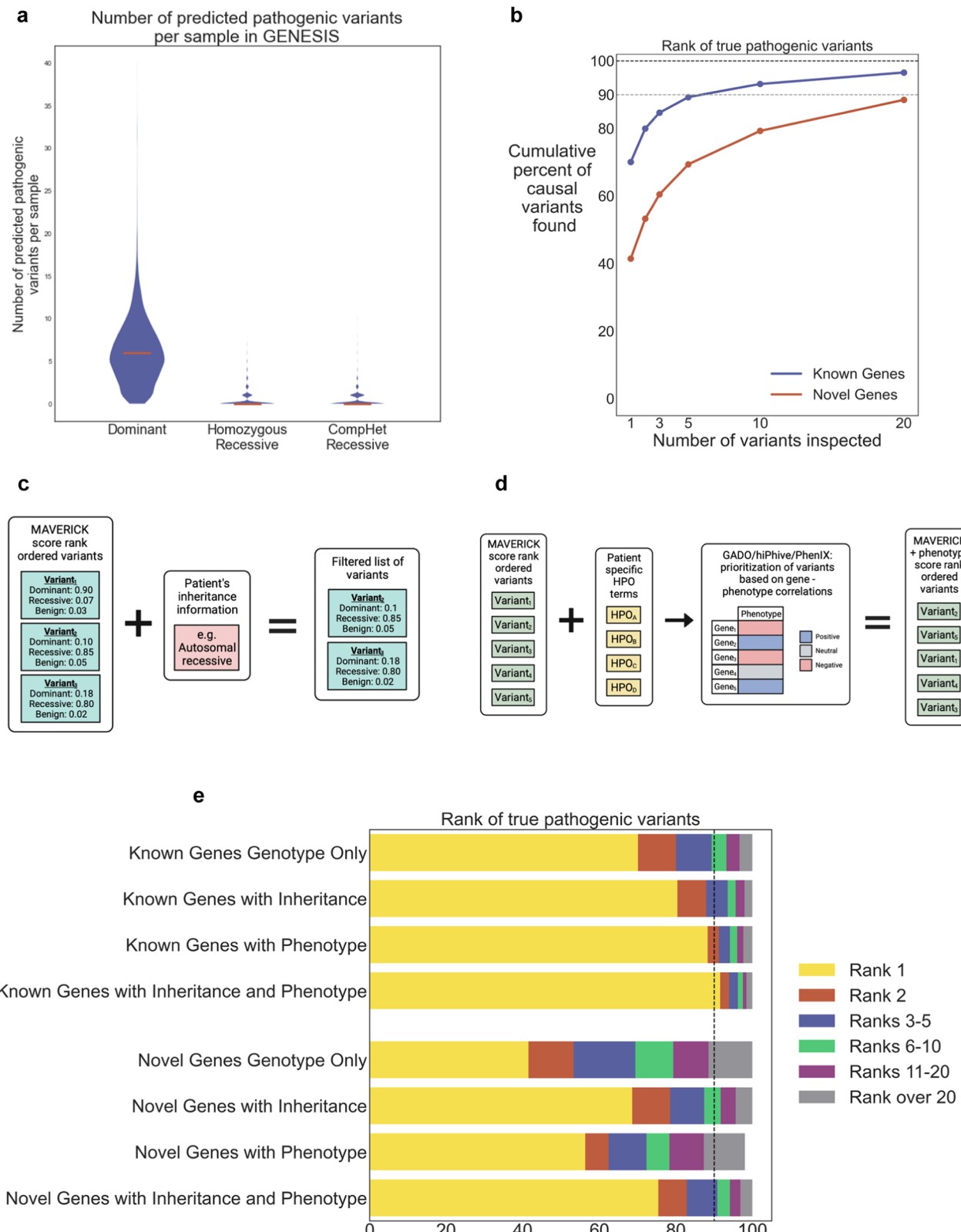

summed the dominant and recessive scores into a single pathogenicity score for each tool in order to demonstrate the effect of that distinction on variant prioritization performance (shown as dashed lines in Fig. 4e, f). Performance of each tool on this task was quantified by calculating the area under the curve of the cumulative percentage of causal variants identified across the simulated cases within the top 20

variants in each case. Simply put, an area under the curve of 1 indicates that all cases were solved on the first guess, while an area of 0 indicates that all cases remained unsolved after 20 guesses per case. Where only genotype information was utilized, MAVERICK outperformed all other tools by a wide margin when the causal variant was a novel variant on a known gene for MAVERICK (Fig. 4e, Supplementary Fig. 6A). The area

**Fig. 3 | MAVERICK reliably prioritizes causal variants in simulated cases.**
**a** Violin plots of the distribution of variants predicted as pathogenic by MAVERICK among 10,138 individuals in the GENESIS database. The median of each distribution is denoted by a horizontal red line. A value of 0.5 was used as the threshold for being 'pathogenic' in this analysis. So, heterozygous variants with a dominant MAVERICK score over 0.5 are shown on the left, homozygous variants with a recessive MAVERICK score over 0.5 are shown in the center, and pairs of heterozygous variants on the same gene with a harmonic mean of recessive MAVERICK scores over 0.5 are shown on the right. **b** Scatterplot of cumulative proportion of cases solved by MAVERICK's rank ordering of variants using genotype information

only when 98 control samples had pathogenic variants from the known and novel genes test sets spiked in. 13,095 pathogenic variants from the known genes set and 696 pathogenic variants from the novel genes set were used. Testing each of these variants in each of the 98 control samples produced 1,351,518 simulated cases. **c** Diagram of how MAVERICK filters variants based on user-provided inheritance information. **d** Diagram of how MAVERICK leverages phenotype information using the GADO, HiPhive, or PhenIX algorithms to adjust its predictions. **e** Stacked horizontal bar plots showing how performance on both the known and novel genes test sets improves when additionally incorporating inheritance and/or phenotypic information along with MAVERICK variant prediction scores.

under the curve was 0.81, while the next best tools were the binary version of MAVERICK with an AUC of 0.69, MAPPIN with an AUC of 0.43 and CADD with an AUC of 0.40 (Supplementary Fig. 6A). Similarly, where the causal variant was on a novel disease gene (Fig. 4f, Supplementary Fig. 6B), MAVERICK outperformed the competition with an AUC of 0.64, while the next best performers were again the binary version of MAVERICK, CADD, and MAPPIN with AUC scores of 0.48, 0.37, and 0.33, respectively (Supplementary Fig. 6B). In each case, the second-best performing tool was the binary version of MAVERICK, suggesting that while the differentiation between dominant and recessive variants improved performance in this task somewhat, it is not the only facet of MAVERICK that allows it to excel at this ranking task.

We further interrogated the relative performance of MAVERICK, MAPPIN, and the 36 non-Mendelian pathogenicity classifiers at this variant ranking task by incorporating inheritance and phenotypic information, similar to what is shown in Fig. 3c–e. MAVERICK maintained top performance, as quantified by the area under the curve, for both the known and novel gene sets in this task where inheritance, phenotype, or both pieces of information were applied to filter and re-prioritize the variants (Supplementary Fig. 6C–H). However, the difference in score between MAVERICK and the next best performing classifier was diminished greatly by incorporating these orthogonal pieces of information, particularly phenotypic information on the cases solved by novel variants on known disease genes.

### MAVERICK reliably prioritizes causal variants in real cases
Encouraged by the strong performance of MAVERICK in the spike-in analyses above, we collected a cohort of 644 patients from the GENESIS database, who had already received a genetic diagnosis and evaluated how many of them MAVERICK could solve. We incorporated 1-3 HPO terms per patient to leverage phenotypic information using the Phenix algorithm[22] and also filtered according to the mode of inheritance. Overall, MAVERICK solved 492 of the 644 cases (76.4%) on the first guess. Furthermore, it solved 615 of the cases (95.5%) within five guesses and it solved 638 of the cases (99.2%) within twenty guesses. Figure 5a–c shows the results of this analysis stratified by whether the causal variant was in MAVERICK's training set (5A, n = 130), a novel variant on a gene represented in the training set (5B, n = 375), or a variant on a disease gene novel to MAVERICK (5C, n = 139). In the case of compound heterozygous pairs, if either were in the training set, then the sample was considered solved using a variant from the training set. The performance on cases where the solution was a novel variant on a known disease gene (Fig. 5b) shows remarkable consistency with the spike-in analysis presented in Fig. 3e. A similar level of consistency is seen for the cases where the solution was on a novel disease gene (Fig. 5c), although not until five guesses. In all, this analysis suggests that the use of MAVERICK as part of a diagnostic pipeline is feasible.

We additionally analyzed this cohort of patients with Exomiser as a point of comparison. A subset of 528 of the 644 patients were used for direct and meaningful comparisons. Supplementary Fig. 7 shows the results of this analysis. The top group is 125 of the 130 patients from Fig. 5a where the causal variant was in MAVERICK's training set and also in Exomiser's ClinVar whitelist. The middle group contains 287 of the

375 patients from Fig. 5b where the causal variant was not in MAVERICK's training set or Exomiser's whitelist, but lies on a gene which is in MAVERICK's training set and has a known gene-phenotype relationship in Phenix. The lower group is 116 of the 139 patients from Fig. 5c where the causal variant neither lies on a gene in MAVERICK's training set nor is in Exomiser's whitelist and does not necessarily have a known gene-phenotype relationship in Phenix. As in the analysis above, we incorporated 1-3 HPO terms per patient to leverage phenotypic information using the Phenix algorithm. Results are shown with and without additionally filtering according to the mode of inheritance. MAVERICK and Exomiser perform largely equivalently on the training/whitelist variants and known genes cohorts when inheritance information is incorporated, though MAVERICK maintains a small advantage when inheritance is not known. MAVERICK outperforms Exomiser on the novel genes cohort both when inheritance information is and is not utilized.

### MAVERICK is able to identify novel Mendelian disease genes
One major goal when creating MAVERICK was for it to be able to solve cases even when the causal variant falls on a yet-to-be-discovered disease gene. Figure 5c shows MAVERICK's performance on a set of cases where the solution fell on a gene that was novel to MAVERICK, however many of those were simply due to idiosyncrasies of how the training set was constructed and those genes were not true novel disease genes. Figure 5d plots MAVERICK's performance at solving 36 cases spanning five recently discovered disease genes, which include both dominant and recessive genes[36–40]. Most, but not all, of these cases are a subset of those shown in Fig. 5c. MAVERICK did not receive inheritance or phenotype information in this analysis. MAVERICK solves 80.6% of these cases on the first guess, 94.4% by the third guess, and all of them within 10 guesses. MAVERICK has further aided in the discovery of *FICD*, a novel Mendelian disease gene for upper and lower motor neuron disease which was recently published[41], as well as several other novel Mendelian disease genes which are in submission for publication.

### MAVERICK outperforms Exomiser on a challenging set of real patients with novel disease genes
Simulating patient phenotypes by picking HPO terms associated with the disease in OMIM or Orphanet obviously oversimplifies issues of patient presentation. For a truer test of MAVERICK's ability to prioritize causal variants on novel disease genes, we used 18 patients that our group had recently solved for novel disease genes (*CADM3, PRDX3, UBAP1, ATP1A1,* and *SORD*) for whom we had physician notes on the presenting symptoms. A physician (author M.F.D.) converted the notes into HPO terms for each patient (range of 1–12 terms per patient). Variants in each individual were scored by MAVERICK and then averaged with HiPhive's score[22] for each gene based on that individual's set of HPO terms. We also evaluated each individual with Exomiser as a point of comparison. Exomiser struggles with this set of patients. Without phenotype information, Exomiser requires a median of 99 guesses to solve these cases. But even with phenotype information, Exomiser still requires a median of 72 guesses to solve these cases. Maverick without phenotype information performs much better than Exomiser, but still not quite as well as it did on the spike-in analyses of

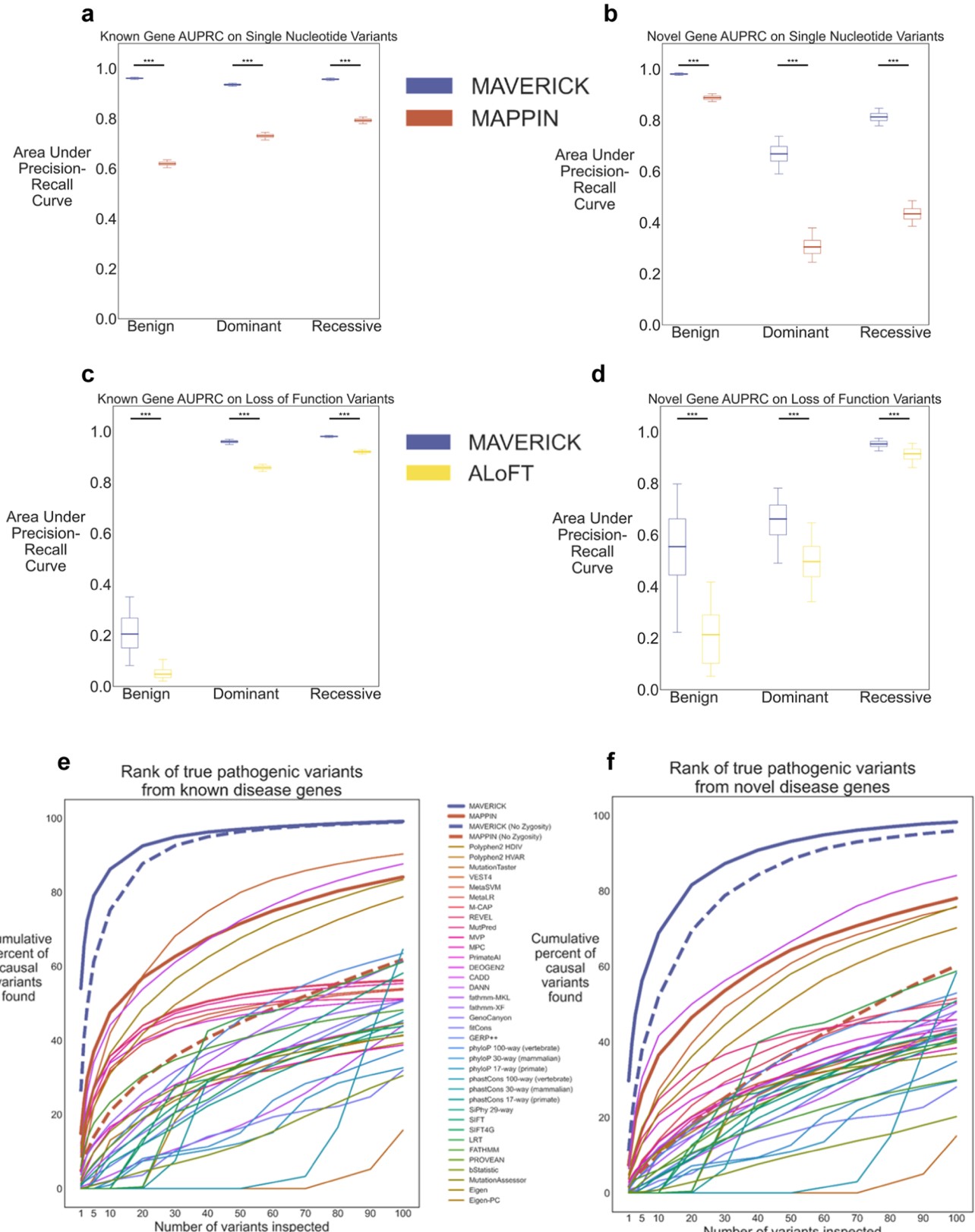

novel genes: a median of seven predictions is required to solve these cases, as compared to a median of three in the spike-in analysis (Fig. 5e, Supplementary Data File 1). When the phenotype information is incorporated, the results improve considerably with a median of three predictions to solve the cases and all but one of the cases being solved within five predictions.

## MAVERICK effectively identifies pathogenic variants on the X chromosome

MAVERICK was trained only on autosomal data and thus far has been evaluated only with autosomal data. We ran MAVERICK on a set of 5244 variants on the X chromosome and found that it still accurately differentiated pathogenic variants from benign ones, but had a

**Fig. 4 | MAVERICK outperforms similar pathogenicity predictors. a, b** Box plots of areas under the precision-recall curve for MAVERICK and MAPPIN missense, stop-gain and stop-loss single nucleotide variants. **a** Performance for the known genes test set. **b** Performance for the novel genes test set. **c, d** Box plots of areas under the precision-recall curve for MAVERICK and ALoFT loss-of-function variants. **c** Performance for the known genes test set. **d** Performance for the novel genes test set. In panels **a–d**, *** indicates *p*-value < 0.001 in one-sided Mann–Whitney U test based on bootstrapping (*n* = 1000 iterations). Multiple testing corrections were not performed. Box plot elements in panels **a–d** show the median as the center line, the 25th and 75th percentiles as limits of the boxes, and the 5th and 95th percentiles as the limits of the whiskers. Outliers are not plotted. **e, f** Line plot of cumulative proportion of cases solved by pathogenicity classifiers when rank ordering variants using only genotype information in SNV-only subset of the spike-in variant prioritization task shown in Fig. 3b. Dashed lines for MAVERICK and MAPPIN show their performance when dominant and recessive scores are summed to give a pathogenicity prediction which does not consider variant zygosity. **e** Performance for the known genes test set. **f** Performance for the novel genes test set.

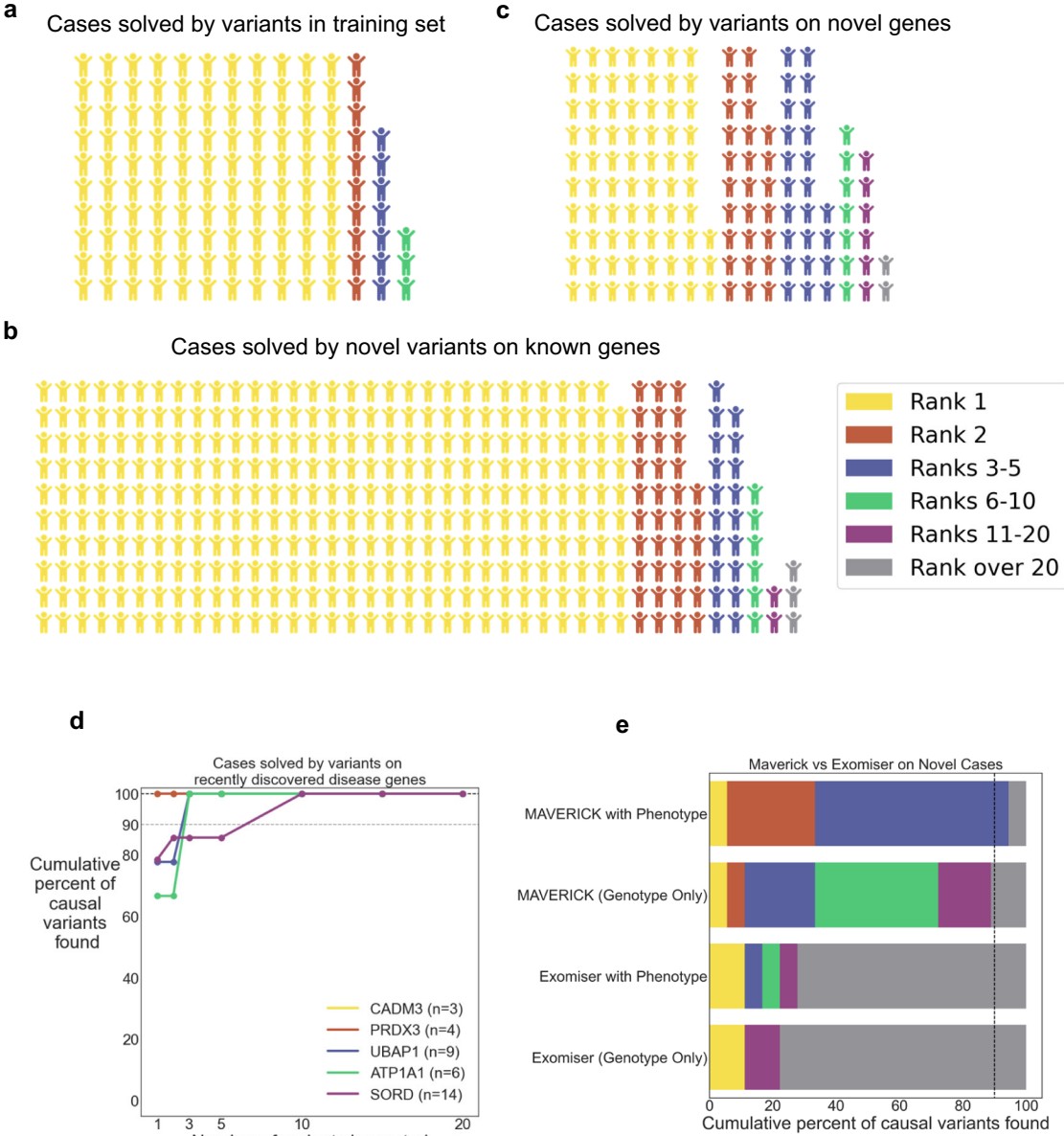

**Fig. 5 | MAVERICK reliably prioritizes causal variants in real cases. a–c** Waffle plots showing rank of causal variant in 644 real cases solved by MAVERICK's rank ordering of variants with inheritance and phenotypic information incorporated. **a** MAVERICK performance on 130 cases whose causal variant was in MAVERICK's training set. Cases where either variant in a compound heterozygous pair are in the training set are included here. **b** MAVERICK performance on 375 cases whose causal variant was not in the training set, but at least one pathogenic variant on that gene was in the training set. **c** MAVERICK performance on 139 cases whose causal variant was on a gene without any pathogenic variants in the training set. **d** Cumulative proportion of real cases solved by MAVERICK's rank ordering of variants for 36 cases whose causal variants were on disease genes that were recently discovered and novel to MAVERICK. The *CADM3* and *PRDX3* cases are all solved on the first guess and so the *CADM3* line is obscured by the *PRDX3* line. **e** Stacked horizontal bar plots of the cumulative proportion of 18 patients with causal variants on novel disease genes solved by MAVERICK or Exomiser's rank ordering of variants. Results are shown using genotype information only as well as with phenotypic information incorporated.

systematic bias of labeling those pathogenic variants as dominant much more than would be expected by random guessing (Supplementary Table 5). Specifically, in the binary classification setting of labeling variants as benign or pathogenic, MAVERICK achieves an auROC of 0.992 and an auPRC of 0.988. Extending this out to the three-class system lowers performance considerably with auPRC for dominant variants being 0.462 and for recessive variants being 0.591. MAVERICK's difficulty resolving dominant from recessive variants on the X chromosome is likely due to the difference in distribution of gene constraint values for the X chromosome relative to the autosomes.

## Discussion

We are presenting a neural network ensemble for Mendelian variant pathogenicity prediction, called MAVERICK. Multi-layered neural network-based learning approaches likely represent a superior approach as suggested by recent success in natural language, protein folding, and image analysis[42–44]. Neural network algorithms were inspired by the human brain and the concept of learning from large amounts of high-fidelity data. For the task of genomic variant categorization, the amount of quality data is limited as it is a function of the genomic space unequivocally known to be linked with high penetrance genomic disease. We chose ClinVar as a truth source for variant curation as it (1) represents one of the largest curated databases; (2) contains independent input from a variety of professional clinical laboratories; and (3) increasingly adopts unified pathogenicity frameworks, especially the ACMG variant prediction guidelines[6,33].

MAVERICK performed highly at identifying ClinVar variants that had been spiked-in to exome samples in order to simulate patients. This performance was largely replicated when assessing a sizeable cohort of real patients. Specifically, in the simulated cases, MAVERICK was able to solve 95% of cases caused by novel variants in known genes within three guesses, when incorporating inheritance and phenotypic information. In the set of 375 real patients diagnosed by novel variants in known genes, MAVERICK similarly solved 95.2% of the cases within three guesses. For novel disease genes, MAVERICK solved within the five highest-ranked variants 90.8% of the simulated cases and 89.2% of the 139 real cases. Within the top 10 ranked variants per exome, the system solved 94.1% and 94.2% of cases. Across our tests, MAVERICK achieves a much lower false positive rate for variants that are benign in the Mendelian sense compared to similar previously published methods. This performance ultimately requires less follow-up of fewer prioritized variants.

When evaluating tools such as MAVERICK, one must also consider that each human genome likely contains multiple alleles with strong genetic effects on traits and diseases, common and rare. Thus, the pathogenic variant causing the phenotype of interest in a specific case might not be ranked highest every time—especially as MAVERICK does not receive phenotypic information in its base evaluations. MAVERICK appears to prioritize variants with the highest potential functional impact when comparing reference to altered protein sequence. It is then up to phenotypic and other considerations that are available to a human evaluator to make final determinations. As the literature and knowledge on genotype-phenotype relationships grows, tools such as MAVERICK will gradually begin to match the accuracy of the best specialists.

MAVERICK performed well in comparison to other Mendelian pathogenicity classifiers. MAPPIN and ALoFT each have a major shortcoming in that they only evaluate a subset of coding variant classes: MAPPIN is limited to SNVs and ALoFT only analyses putative loss-of-function variants. More problematic, however, is the high false positive rate introduced by MAPPIN and ALoFT compared to MAVERICK. On both the known genes and novel genes test sets, both MAPPIN and ALoFT had a lower precision than MAVERICK for the dominant and recessive classes, often by a wide margin

(Supplementary Tables 3 and 4). This would make using MAPPIN and ALoFT for prioritizing causal variants in patient samples challenging as the large excess of benign variation will cause a large number of false positives, which will obscure the true causal pathogenic variant. This is the reason that precision and recall are used as the major performance metrics throughout the paper: precision in an unbalanced dataset gives an indication of how difficult it is to spot a pathogenic variant among many benign ones; and recall reveals what proportion of the pathogenic variants would be identifiable in this way. While far from comprehensively testing these ideas, the precision and recall metrics are ultimately much more informative in these scenarios than metrics such as accuracy or specificity. When MAVERICK's performance is compared to other pathogenicity classifiers in the variant prioritization task, we see that it outperforms all other methods, including MAPPIN. MAVERICK's advantage over the other programs is largest when only genotype information is utilized and can be mitigated by the inclusion of inheritance and phenotypic information. Finally, MAVERICK is conceptually most similar to Exomiser[22]. Exomiser uses a combination of variant pathogenicity prediction, gene-level phenotype association, and variant pathogenicity whitelists to rank variants within an individual. Exomiser performs highly when the causal variant in a known disease gene is already categorized in ClinVar. It still performs well in evaluating novel variants in known disease genes, thanks to its incorporation of phenotypic information. Where Exomiser predictions are weakest is in categorizing variants in novel disease genes. In Fig. 5e and Supplementary Data File 1, we show that MAVERICK's genotype-only approach strongly outperforms Exomiser in such cases of novel variation. In this example of 18 cases, the phenotype information provided to each tool resulted in a significant increase in performance. Since phenotype prioritization tools like Exomiser's HiPhive rely mostly on animal models to establish gene-phenotype relationships, this boost in performance was likely due to de-prioritization of variants on genes that were clearly not related to the phenotype rather than active prioritization of the causal gene. HiPhive likely will improve coverage of genes and their associated phenotypes due to the continued creation and study of animal models.

Since November of 2020, evolving MAVERICK predictions have been tested by users of the GENESIS genome platform which contains >17,000 datasets. Due to this long assessment period, we have received extensive feedback. Several novel disease genes were identified using GENESIS in this period with the demonstrated support of MAVERICK as it acted as a computer-aided genomics tool. Noted limitations of the system include the lack of consideration of specific variant classes, such as splice-altering variants, non-coding variants, and structural variants, as well as a preference to rank strong (Mendelian) allelic effect sizes high and variants implicated in increased risk for complex genetic diseases as benign. False positive predictions may be increased in samples with a low quality of variant calls. A further limitation of MAVERICK is its strict reliance on the dominant, recessive, and benign categorization of variants. While this scheme provides benefits in the context of identifying causative pathogenic variants in individuals with rare, monogenic diseases, scenarios involving incomplete penetrance, reduced expressivity, and even co-dominance of alleles are viewed as exceptions in MAVERICK's conceptualization rather than part of a continuous spectrum of variant effects, of which Mendelian effects are only a part. Overcoming this limitation should be a major focus of future pathogenicity prediction algorithm development efforts.

In summary, we have shown that MAVERICK identifies Mendelian pathogenic variants more accurately and with a broader scope than any other tool tested. This is achieved in a deep learning approach rather than a predetermined framework of rules. Notably, the immediate amino acid neighborhood of a variant is a major unbiased driver of this method. MAVERICK has a low false positive rate that enables a reliable identification of the causal variant in patients with monogenic diseases. In the clinical setting this will lead to higher

diagnostic efficiencies. In genetic research, MAVERICK is already fostering gene identification. We envision that further improvements will include additional variant classes, integration with established variant classification frameworks and heuristic methods, and higher precision and recall. This will ultimately lead to a high degree of cooperation of computer-aided genomics tools with human genomics specialists.

## Methods

### Creation of training, validation, and test sets of variants

The January 2020 variant summary report was downloaded from ClinVar and used as the primary basis for the training and validation sets. The version of the OMIM database from January 14, 2020 was downloaded as well. The ClinVar dataset was filtered to identify germline variants with criteria provided and no conflicts in interpretation of pathogenicity (one star or higher). From that set, we selected the following variant types: single nucleotide variants, deletions, duplications, insertions, indels, and microsatellites. We further selected only the variants that were annotated as benign, likely benign, benign/likely benign, pathogenic, likely pathogenic, or pathogenic/likely pathogenic. In order to interpret pathogenic variants as dominant or recessive and select for only Mendelian variants, we further selected only the entries that cited an OMIM phenotype identifier. We used our downloaded version of the OMIM database to map these phenotype identifiers to patterns of inheritance. Most phenotypes in OMIM have only one mode of inheritance (even if the gene has multiple modes of inheritance). Among the cases in which the variant was mapped to a phenotype with both dominant and recessive inheritance annotated, we removed that variant from the set. Additionally, we removed any variants in which the associated OMIM identifier given in ClinVar was deemed to be incorrect. For example, we found entries in which the OMIM identifier pointed to a different gene than the variant was on. We also excluded OMIM terms that were annotated as 'non-diseases', 'susceptibility', or 'putative' (brackets, braces, or question marks). We also included any variants found in gnomAD v2.1.1 in the homozygous state in at least two individuals as benign variants.

To select down to the non-splicing protein-altering variants within this set, we utilized Annovar (version 2018-04-16)[45]. We worked with the GRCh37 coordinates for each variant and employed the Gencode V33 Basic annotation of the human genome (lifted over to GRCh37 coordinates)[46]. We used Annovar's annotate_variation.pl and coding_change.pl scripts to identify the protein-sequence changes caused by each variant on each isoform of each gene they affect. This also provided us with information on variants near splice sites and we chose to remove all variants within 2 bp of canonical splice sites. Most protein-altering variants were then seen to affect more than one transcript of the affected gene. We selected a single transcript for each variant as follows: if the variant affects the canonical transcript of the gene (according to gnomAD's definition of canonical transcripts), then use the canonical transcript; if none of the affected transcripts are the canonical one, then use whichever has the highest expression across tissue types in GTEx V7 (using the median of samples as the representative for each tissue type); if multiple genes are affected by this variant, pick the gene whose canonical transcript is affected; if multiple genes have their canonical transcripts affected by this variant, pick the gene whose average expression is highest across tissue types in GTEx V7. In this way, we select down to a single amino acid sequence and how it is altered for each variant.

Next, we collected several numerical annotations for each variant, which are served as structured information to the MAVERICK model. For each variant, we collected the allele frequency and number of times seen as a homozygote among controls in gnomAD v2.1.1[8]; the gnomAD constraint information for the canonical transcript of the gene (regardless of whether the variant affected the canonical transcript or not) in the form of the probability that transcript is loss-of-function intolerant (pLI), probability that transcript falls into distribution of recessive genes (pRec), the probability that transcript falls into distribution of unconstrained genes (pNull), Z-score for missense variants in gene, and Z-score for loss-of-function variants in gene[8]; the pext score from gnomAD[27]; the local constraint score (CCR) for the affected residue[31]; the gene damage index (GDI) score for the associated gene[29]; the RVIS score for the associated gene[30]; and the GERP++ score for the nucleotide harboring the variant[28]. For residue-level scores (CCR, pext, and GERP) on deletions that span multiple residues, we used the maximum score within the affected span. For residue-level scores on insertions, we use the maximum score of the two neighboring positions. These sources of structured information were chosen in an effort to supply useful information to the MAVERICK model while minimizing the risk of propagating circularity. As such, these scores were selected because they are based on observations that should be relatively uniform in quality across all genes, regardless of how well-studied a gene is.

The final annotation that we created was the evolutionary conservation track for each gene transcript. This approach was modeled after the procedure used to generate input for NetSurf-P2[47]. For each protein-coding transcript in the Gencode V33 Basic annotation of the GRCh37 genome, MMSeqs2 (Release 11) was used to generate multiple sequence alignments against the August 2018 version of Uniclust90[26,48]. This was a two-step process, first "mmseqs search" was run with "num-iterations" set to 2 and "max-seqs" set to 2000. Second, "mmseqs results2msa" was run with the default parameters using the output of the first step. These multiple sequence alignments were then run through HHSuite's hhmake utility[49] using default settings except with the parameter "-M" set to "first". Finally, these HHM profiles were parsed into compressed NumPy arrays for easy loading as input to the model.

The protein-altering variants that passed the filtering procedure detailed above were annotated with the appropriate residue-level, transcript-level, and gene-level structured information. The amino acid sequences of the reference and altered protein were saved for each variant as well. This yielded 126,739 variants. One thousand of those variants were randomly selected to serve as the validation set. This contained 778 benign, 108 pathogenic or likely pathogenic dominant, and 114 pathogenic or likely pathogenic recessive variants. The remaining variants composed the training set, which had 99,380 benign, 13,112 pathogenic or likely pathogenic dominant, and 13,247 pathogenic or likely pathogenic recessive variants. The validation set was used for hyperparameter tuning and model selection and as a result, MAVERICK's performance on that set is slightly better than would be expected in general. Therefore, we did not utilize it for primary evaluations.

In order to construct the known and novel genes test sets, we downloaded the January 2021 variant summary report from ClinVar and repeated the above procedure, but at the end removed the 126,739 variants that were already in the training and validation sets. There were 17,942 variants that passed these filters. These could have been newly added to ClinVar, upgraded from zero star to a higher rating, or had an OMIM phenotype term associated with them. The variant set was then split into the 16,012 that fell on genes that had at least one pathogenic variant in the training set (the known genes set) and the 1930 that fell on genes that had no pathogenic variants in the training set (the novel genes set). The known genes set contained 2917 benign, 6085 dominant, and 7010 recessive variants. The novel genes set contained 1234 benign, 183 dominant, and 513 recessive variants.

### MAVERICK architecture

MAVERICK is an ensemble of eight models spanning two distinct architectures. This combination was used in order to encourage a diversity in the types of evidence used by the ensemble members and thereby to increase the likelihood of training a well-calibrated final model. The ensemble is also more accurate overall than any of its individual members.

Architecture 1 uses a tandem stack of transformers to process the conservation information and then runs the outputs of those transformers, along with the structured information through a classification head which returns the likelihood that each variant is benign, dominant, or recessive. This architecture is depicted in Supplementary Fig. 1. Specifically, for each variant, the evolutionary conservation track for the reference version of the protein is taken and the 100 amino acids upstream and downstream of the site of variation (so, 201 amino acid span in total) is extracted. This 201 amino acid span of the reference protein sequence along with its conservation is the first input. The variation is then applied to this reference protein sequence. In the case of missense variants, this is simply changing the particular amino acid, while leaving the conservation information unchanged. For deletions, this involves marking that particular residues are omitted: a row in the input matrix has values of 1 for each entry and so the deleted residue(s) are then set to a value of 0. For insertions, this involves adding in new amino acids that have no conservation information. Specifically, new columns are inserted into the conservation matrix with 0 s for all conservation values and the new amino acid encoded. For as many residues are inserted, as many are removed from the end of the sequence to maintain a span of 201 amino acids (the first 100 amino acids always remain the same). For frameshifts, the amino acids are changed all the way until a stop codon is reached, with the conservation information left unchanged. Residues following the novel stop codon are 'logically deleted' in the same way as described for deletions, thereby maintaining information about the conservation of the original sequence. Similarly for stop-gains, the residues after that point are marked as omitted while the conservation track remains unchanged. This altered protein sequence with conservation information is the second input to the model. The final input to the model is the structured information.

The neural network model is written in Tensorflow using the Keras API. First, the reference and altered protein sequences with conservation information are projected through a standard Dense layer using Einsum operations to alter the dimensionality of the inputs from 51 to 64 in order to facilitate computations through the transformer layers. The input sequences are then masked and have position embeddings added to them in the standard fashion for natural language processing work. The mask tells the transformers to ignore entries in the input matrices where all rows have values of 0. This happens when a protein is shorter than 200 amino acids, when a variant is within 100 amino acids of the start or within 100 amino acids of the end of the protein sequence. The position embedding adds a unique representation of each position within the sequence to each residue using sine and cosine functions of varying frequencies. This is done so that the transformers can understand the order of the residues.

The reference and altered protein sequences with conservation are run through the tandem stack of six transformer layers. Each transformer uses 16 attention heads and an intermediate dense layer size of 256. The stacks that process each input have shared weights. The outputs of each stack are then sliced to only use the center token (which is the site of variation) and passed through a dense layer. This produces a fixed-length embedding of 64 values for the reference and the altered sequence inputs. The embedding of the reference sequence is then subtracted from the embedding of the altered sequence. This difference in their embeddings is then concatenated with the embedding of the altered sequence.

Meanwhile, the structured information is normalized by a quantile transformer fit to the distributions within the training dataset and processed using a Dense layer of size 64. The output of this layer is then concatenated to the embeddings of the altered sequence and the difference of the reference and altered sequences. These three components are each of size 64 when they are concatenated together, producing a representation of size 192. This integrated representation is then passed through a three-layer classification head of Dense layers

of sizes 512, then 64, and then 3 to produce the final classification output.

There are three models in the ensemble based on architecture 1. One of the models was trained with default class weights, while the other two used class weights to add more importance to the dominant and recessive classes. Specifically, they gave recessive variants a weight of 7, dominant variants a weight of 2, and kept benign variants at a weight of 1. This means that, during training, each recessive variant contributes 7 times as much toward the loss function as a benign variant, while a dominant variant contributes twice as much as a benign variant. This puts more emphasis on classifying recessive and dominant variants correctly in the loss function. Models based on architecture 1 each have 473,539 trainable parameters.

Architecture 2 is depicted in Supplementary Fig. 2. It also takes three inputs, but does not use the reference protein sequence with conservation. Instead, it takes as its third input a 201 amino acid span of the altered protein sequence, centered on the site of variation (no conservation information). This protein sequence is run through ProtTrans' ProtT5-XL-BFD as a feature extractor. ProtT5-XL-BFD is a protein language model trained to predict masked amino acid residues from millions of protein sequences. As a result, its representations of protein sequences have information about each residue's secondary structure and solvent accessibility, as well as protein-level features like subcellular localization. The alternate protein sequence is converted to a dense embedding by ProtT5-XL-BFD and this is then passed through a bidirectional long short-term memory (LSTM) layer, the final internal state of which is used as a fixed-length representation of the sequence of size 64. The alternate protein sequence with conservation information is processed exactly as in architecture 1, through a six-layer stack of transformers, then has its center token sliced out and passed through a dense layer to generate a fixed-length representation of that input also of size 64. The structured information is processed exactly as in architecture 1, to a size of 64. The fixed-length representations of the altered protein sequence and the altered protein sequence with conservation information are then concatenated together with the structured information, producing a representation of size 192. This integrated representation is then passed through a three-layer classification head of Dense layers of sizes 512, then 64, and then 3 to produce the final classification output.

There are five models in the ensemble based on architecture 2. Three of the models were trained with default class weights. The fourth model gave dominant variants a weight of 2 and recessive variants a weight of 3. The fifth model gave dominant variants a weight of 2 and recessive variants a weight of 7. Models based on architecture 2 each have 723,651 trainable parameters.

All models were trained with a batch size of 128 for 20 epochs using SGD as the optimizer. The learning rate and momentum were cycled by a OneCyclePolicy[50] with a maximum learning rate of 0.1, an initial learning rate of 0.001, and an initial momentum of 0.95. Briefly, this method adjusts the learning rate and momentum of the SGD optimizer over the course of the 20 epochs so that learning rate ramps up to 0.1 from 0.001 over the first 30% of the training steps. During this time, momentum ramps down from 0.95 to 0.85. Over the latter 70% of the training steps, the learning rate is decreased from 0.1 to 0.0000004 while the momentum is decreased from 0.95 to 0.85. The major hyperparameters that were tuned are cataloged in Supplementary Table 6. The categorical cross-entropy of the validation set was used to select the best models for the ensemble.

## Ablation experiments

The ablation experiments were conducted by adding dropout layers to the models between the relevant input layers and the first layer within the model to which they originally connected. These dropout layers were then coerced to drop all of the connections between their input and the subsequent layer during inference. In this way, individual

inputs or any combinations of inputs could be dropped from any ensemble member as desired. These experiments were done only on the trained MAVERICK ensemble members rather than by re-training the models without the ablated input(s).

## Cross-validation experiments

The genes in the training set were divided into the 1930 which contained pathogenic variants and the 13,219 that contained only benign variants. The variants from 386 (20%) of the genes containing pathogenic variants and 2643–2644 (20%) of the genes containing only benign variants were then placed into each of five data folds so that each fold contained all variants from 20% of the genes. Since genes differ in their numbers of variants, the number of variants in each fold ranged from 23,283 to 27,496 even though the number of genes in each fold was always either 4029 or 4030. For each of the eight submodels of the MAVERICK ensemble, five-fold cross-validation was then performed using these training folds and performance of the model was evaluated on the held-out fold. This resulted in the creation of 40 models which comprise CV-MAVERICK. No hyperparameter tuning was performed during this process. All models were trained identically to their corresponding MAVERICK ensemble component except that the training was performed on only 80% of the data and so was run for 20% fewer training steps. Performance of CV-MAVERICK on the known genes and novel genes test sets was calculated by averaging the predictions of each of the 40 models on those test sets.

## Spike-in analyses

Ninety-eight patients with rare, Mendelian diseases were chosen from whom whole exome data was available that had yielded a genetic diagnosis. We re-processed the exome datasets according to GATK4 best practices and called variants[51]. Briefly, samples were aligned to GRCh37 using BWA[52], duplicate reads were marked with Picard's MarkDuplicates utility, base quality score recalibration was performed with GATK 4.0. GATK's HaplotypeCaller was then run to generate gVCF files for each sample, which were then resolved into variant calls with GATK's GenotypeGVCFs utility. We then manually removed the variant call (or pair of calls) from each sample that had been identified as the causal pathogenic variant in order to make these patients pseudo-controls. We chose this method because initial tests using samples from the 1000Genomes cohort proved to be too easy since almost all of their variants are present in gnomAD, which MAVERICK takes as a strong indication that the variants are not dominant pathogenic, thus providing an unfair advantage. We further filtered the samples to only include high-quality variant calls using GATK's CNN filter at default settings as well as requiring a read depth of 20 for each variant and that for heterozygous variants there be at least half as many reads supporting the alternate allele as there are supporting the reference allele. All remaining protein-altering variants were then scored with MAVERICK. To create a rank ordering of the variants in each sample, we created a final score. For heterozygous variants, MAVERICK's predicted dominant score was used as this final score. For homozygous variants, MAVERICK's predicted recessive score was used as this final score. Additionally, compound heterozygous pairs were generated by taking the harmonic mean of the recessive score for each pair of heterozygous variants on each gene. This set was filtered so that two variants could not be a compound heterozygous pair if HaplotypeCaller had determined they were part of the same haplotype. All variants and compound heterozygous pairs of variants for each sample were then sorted according to this 'final score'.

For the spike-in analyses, dominant variants from the test sets were placed into each sample to determine where their dominant score would rank among the 'final scores' for all of that sample's variants. Similarly, recessive variants were ranked in the same way but using their recessive score. We recognize that this means all spiked-in recessive variants were tested as if they were seen as homozygotes. This may appear to be an advantage, but it actually is not. Since the compound heterozygous pairs are scored according to their harmonic mean, the score for a pair of identical heterozygous variants would be numerically equal to that of a homozygous variant. So we believe that testing the recessive variants as homozygotes does not provide any unfair advantage and makes the results simpler to interpret.

## Ensemble components analysis

While conducting the spike-in analyses of MAVERICK's performance on the known genes and novel genes test sets, the individual performance of each of the eight ensemble components was also collected. Areas under the curve for these analyses were calculated using scikit-learn's implementation of the AUC metric where the curve plots the cumulative proportion of samples solved by the top-k guesses for the top 20 ranked variants in each simulated individual for each submodel. The data are normalized so an area under the curve of 1 would correspond to a model solving every sample on the first guess.

## Simulation of patient phenotypes

In order to simulate patient phenotypes for each variant in the known genes and novel genes sets, we exploited the fact that each variant was associated with an OMIM phenotype term due to the manner in which the training and test sets were created. We then used the HPO annotation (downloaded June 21, 2021) to find the HPO terms associated with each OMIM phenotype. If there were more than five HPO terms associated with any OMIM phenotype, five were randomly selected.

## Scoring phenotypes with GADO, HiPhive, and Phenix

Scoring phenotypes with HiPhive and Phenix was accomplished using the Exomiser REST Prioritiser version 12.1.0. Exomiser data version 2003 (from March of 2020) was used so that the performance of HiPhive and Phenix could be accurately assessed on known and novel genes without the passage of time giving them unfair knowledge of the novel disease genes. The list of up to five HPO terms was passed to the REST prioritiser for each variant, which returns scores between 0 and 1 for every gene. For genes not in Exomiser's annotation set (and therefore without a score), we assigned a phenotype score of 0.5.

Scoring phenotypes with GADO required first converting the set of HPO terms for the OMIM phenotype into the lowest parent term on the HPO graph that was scored by the GADO method. These are referred to as the significant HPO terms. We selected a maximum of five significant HPO terms for each OMIM phenotype. Next, we downloaded the GADO prediction matrix of Z scores from https://molgenis26.gcc.rug.nl/downloads/genenetwork/v2.1/genenetwork_gene_pathway_scores.zip. As described in the GADO paper, known associations between genes and HPO terms were then set to a value of 3. To convert the data from the Z-score range to a more useful range for our purposes, we applied a sigmoid function to compress the scores to a range of 0 to 1. The distribution of values in this matrix was centered on 0.5 and any genes without entries in this matrix were also assigned phenotype scores of 0.5. To compute gene-phenotype scores for sets of 'significant' HPO terms, we took the arithmetic mean of the values of the individual gene-phenotype scores from this matrix.

To combine the phenotype scores from GADO, HiPhive, or Phenix with the MAVERICK score, we took the arithmetic mean of the appropriate MAVERICK score (the 'final score' described above) and the phenotype score for that variant's gene according to each of these tools.

## Comparison to other pathogenicity classifiers

Predictions for all protein-altering SNVs by MAPPIN for hg19 were downloaded from https://doi.org/10.6084/m9.figshare.4639789. Predictions for all premature stop variants by ALoFT for hg19 were downloaded from https://aloft.gersteinlab.org. Predictions for all missense SNVs by all other tools for hg19 were downloaded from dbNSFP v4.0.

MAVERICK was run on the known genes and novel genes test sets to predict the scores for those variant sets. The subset of variant types appropriate for each tool (missense and stop-gain for MAPPIN, and stop-gain and frameshift for ALoFT) were then selected and the scores for those variants were extracted from the pre-calculated lists above. We were not able to get the command line version of ALoFT to install, limiting us to only assess stop-gain SNVs for it.

For the spike-in analysis, we collected the predictions by MAVERICK, MAPPIN, Polyphen2, MutationTaster, VEST4, MetaSVM, MetaLR, M-CAP, REVEL, MutPred, MVP, MPC, PrimateAI, DEOGEN2, CADD, DANN, fathmm-MKL, fathmm-XF, GenoCanyon, fitCons, GERP++, phyloP, phastCons, SiPhy, SIFT, SIFT4G, LRT, FATHMM, PROVEAN, MutationAssessor, and Eigen for each SNV in the known and novel genes test sets along with each SNV observed in the control individuals. We additionally summed the dominant and recessive scores for MAVERICK and MAPPIN to generate their noZygosity overall pathogenicity scores. From this set of predicted scores, the spike-in analysis was carried out as described above, with and without the inclusion of phenotypic and inheritance information.

Areas under the curve for the spike-in analyses were calculated using scikit-learn's implementation of the AUC metric where the curve plots the cumulative proportion of samples solved by the top-k guesses for the top 20 ranked variants in each simulated individual for each tool. The data are normalized so an area under the curve of 1 would correspond to a model solving every sample on the first guess.

## GENESIS

GENESIS is a web-based genomic data management platform designed to facilitate matchmaking among physicians whose rare monogenic disease patients carry identical variants or who carry pathogenic variants on the same gene. It has thus far been involved in the discovery of over 70 novel disease genes. We have built MAVERICK into GENESIS's annotation engine so that all applicable variants are given a MAVERICK score when they are brought into the database. Prioritization of variants in a patient by MAVERICK score is now part of the default sorting scheme. GENESIS also provides numerous orthogonal approaches to prioritize variants, including the incorporation of inheritance information and robust options for filtering variants by their call quality.

## Patient cohort

A cohort of 644 patients with genetic diagnoses for rare monogenic diseases was downloaded from GENESIS. Variants had been called on these patients through a variety of methods including but not limited to GATK and Freebayes. Not all samples had all the same quality checking measures for their variants. Variants were filtered using GATK's Convolutional Neural Network (CNN) filter when available at default settings as well as requiring a read depth of 20 for each variant and that for heterozygous variants there be at least half as many reads supporting the alternate allele as there are supporting the reference allele. Variants seen in more than 1% of the overall population in either gnomAD v2 or v3 were removed. Variants seen in more than 1% of samples within GENESIS were also removed. Variants called with high quality in unaffected samples in GENESIS were also blacklisted out in this analysis if they appeared with the same zygosity or in the same compound heterozygous pair. Variants were filtered to only include those that matched the observed inheritance pattern of the causal variant. GENESIS contains high-level descriptions of the phenotype for most patients which generally takes the form of an ORPHAnet code for a large family of rare disorders. We used these codes to select 1-3 HPO terms for each disease. These were generally very vague with the most common assignment being Peripheral Neuropathy (HP:0009830). The HPO codes were scored using Phenix to generate a gene-phenotype score which was then averaged with the MAVERICK score of each variant on each gene. Ranking was performed as described for the spike-in analyses.

## Comparison to Exomiser on 528 patients

The cohort of 644 patients described above were also used for a comparison with Exomiser, but the cohort was filtered in an effort to make the comparison as direct as possible. Of the 130 patients whose disease was caused by variants in MAVERICK's training set above, 125 were retained which were also in Exomiser's whitelist of ClinVar variants. Conceptually, this allows both tools to recognize the causal pathogenic variant in those individuals. This was referred to as the training/whitelist variants set. Of the 375 patients whose disease was caused by novel variants on genes with other pathogenic variants in MAVERICK's training set, 287 were retained which also were not in Exomiser's whitelist of ClinVar variants but still had a gene-phenotype relationship in Phenix. This was referred to as the known genes set. Of the 139 patients whose disease was caused by variants on disease genes novel to MAVERICK, 116 were retained which were also not in Exomiser's whitelist of ClinVar variants. This was referred to as the novel genes set.

The VCF file for each patient and their associated HPO terms were then submitted to Exomiser running data version 2003. The VCF files were run through MAVERICK and the accompanying HPO terms were scored by Phenix as described in the section above. The Exomiser scores at the gene level were collected for the autosomal dominant and autosomal recessive variants. The scores were sorted by the EXOMISER_GENE_COMBINED_SCORE column to calculate the rank of the causal variant. In comparisons where the inheritance was known, only the appropriate dominant or recessive variants file was used. In comparisons where the inheritance was not known, the two files were combined and sorted before calculating the rank of the causal variant.

## Comparison to Exomiser on 18 patients with clinical notes

To test how MAVERICK prioritizes variants in real patients, we used the Genesis database to select 18 patients with inherited neuropathies recently found to be caused by variants on novel disease genes. In this real-life setting, pseudonymized phenotype information including patient history, clinical examination results, and nerve conduction studies were retrieved from the database of the rare disease clinical research network (RDCRN) for each individual. Between one and 12 HPO terms were assigned per individual, depending on the amount and specificity of available data. This assignment was done by a trained neurologist experienced in the diagnostic procedures of neuromuscular diseases (author MFD). Supplementary Data File 1 lists the HPO terms assigned to each individual, along with the final rankings of that individual's causal variant(s) by MAVERICK and Exomiser each with and without incorporation of phenotypic information.

The whole exome sequencing data from the 18 patients were processed according to GATK best practices as described above for the samples used for the spike-in analyses. Slightly more relaxed quality filtering was used with a depth requirement of only 15 reads and only a quarter as many reads being required to support an alternate allele as support the reference allele. The CNN filter was still used at default settings.

The VCF file for each patient and their associated HPO terms were then submitted to Exomiser running data version 2003. The VCF files were run through MAVERICK and the accompanying HPO terms were scored by HiPhive as described in the section above. The Exomiser scores at the gene level were collected for the autosomal dominant and autosomal recessive variants. The scores were sorted by the EXOMISER_GENE_COMBINED_SCORE column to calculate the rank of the causal variant when leveraging phenotype information and sorted by the EXOMISER_GENE_VARIANT_SCORE to calculate performance when ignoring phenotype information. Exomiser creates these gene-level scores by summarizing each gene as the most likely pathogenic variant or pair of variants in it. The fact that Exomiser is operating at the gene level does give it an advantage in this comparison, but we decided to

run it this way since that is the way most people use Exomiser's output. MAVERICK performance was calculated by ranking variants (with or without the influence of phenotype information) according to a final score calculated as described for the spike-in analyses.

## Calculation of precision, recall, and F1 scores

Precision was calculated as the number of predicted true positives (TP) divided by the total number of predicted positives – true positives plus false positives (FP):

$$\text{Precision} = \frac{TP}{FP + TP} \tag{1}$$

Recall was calculated as the number of predicted true positives divided by the total number of positives – true positives plus false negatives (FN):

$$\text{Recall} = \frac{TP}{FN + TP} \tag{2}$$

F1 score is the harmonic mean of the precision and recall. It was calculated as:

$$F1 = 2\frac{\text{Precision} \times \text{Recall}}{\text{Precision} + \text{Recall}} = \frac{TP}{TP + \frac{1}{2}(FP + FN)} \tag{3}$$

## Reporting summary

Further information on research design is available in the Nature Portfolio Reporting Summary linked to this article.

## Data availability

The pre-computed scores for all missense and nonsense SNVs in Gencode Basic V33 on GRCh37 and lifted over to GRCh38 have been deposited to Zenodo under the DOI 10.5281/ZENODO.7838659[53]. The training, validation, known genes, and novel genes sets have also been deposited in that repository under the same DOI. ClinVar summary reports are available at ClinVar. OMIM gene-phenotype-inheritance reports are available through OMIM. Gene-phenotype association scores produced by Phenix and HiPhive are available through use of the Exomiser software[22]. The GADO gene-phenotype scoring matrix is available at https://molgenis26.gcc.rug.nl/downloads/genenetwork/v2.1/genenetwork_gene_pathway_scores.zip. Raw and processed whole exome data as well as corresponding phenotype data are available through GENESIS. The exome data is subject to controlled access due to the nature of the consent forms signed by the individuals. As a result, sharing this data will require approval by the RDCRC-Inherited Neuropathy Consortium and/or individual investigators and may take several weeks. Contact the corresponding author to initiate this process.

## Code availability

The code for MAVERICK is provided under an MIT open-source license in Supplementary Software 1 as well as at our Github: https://github.com/ZuchnerLab/Maverick[54]. A Colab notebook version of MAVERICK that accepts a VCF file aligned to GRCh37 or GRCh38 as input and returns scores for all applicable variants is linked at the Github. Python notebooks and CoLabs demonstrating the creation of the training and test datasets, as well as to replicate the training process are also given there.

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

## Acknowledgements

We thank the patients and their families who have shared their genetic data with us, without whom this work would not have been possible. We thank the members of the Zuchner lab for their support and insightful feedback throughout this project. This work was supported by the NIH National Institutes of Neurological Disorders and Stroke (grant 2R01NS072248-11A1 to S.Z.). M.F.D. has received funding by the German Research Foundation (Deutsche Forschungsgemeinschaft, DFG, DO 2386/1-1).

## Author contributions

M.C.D. and S.Z. designed the study and wrote and edited the manuscript.

## Competing interests

The authors declare no competing interests.
