## [Peer Review File · Nature Communications]

Deep structured learning for variant prioritization in Mendelian diseasesREVIEWER COMMENTS

Reviewer #1 (Remarks to the Author):

In this manuscript, the authors present MAVERICK, a new method for variant prioritization in Mendelian disease.

The method uses deep learning based on transformer architectures incorporating various information such as amino acid sequences, allele frequencies, gene constraints etc.

As a general comment, I think the methodology part is sound, using a reasonably constructed network architectures.

However, I have concerns on the evaluation part, especially on the comparison with Exomiser, a defacto standard tool for the same task.

Comment 1:

The comparison with Exomiser uses the dataset of only 18 patients obtained by the authors' group.

I have a concern that these patients may be cherry picked for MAVERICK to outperform Exomiser.

The author should additionally perform a benchmark using a larger dataset broadly collected from public databases, as currently done for other tools like MAPPIN, ALoFT, and PrimateAI.

While the authors avoid this by saying "obviously oversimplifies issues", I am not so convinced.

Comment 2:

The problem setting of MAVERICK is strict.

It focuses on Mendelian disease rather than polygenic disease,

uses genotype information rather phenotype information (though optional),

and predicts the mode of inheritance rather than benign/pathogenic classification.

This strictness makes MAVERICK only comparable with few tools, eliminating many conventional tools for variant prioritization.

Thus, the authors should explain more why this problem setting is useful.

For example, is it reasonable to assume Mendelian disease and phenotype information is unavailable etc?

Comment 3:

The authors should avoid expressions like "excellent performance" when describing benchmark results without competitor tools (e.g. Fig 2).

In my opinion, what we can only say in the evaluation of a computational tool is "better or worse" compared to other tools, not absolute "good or bad".

Without competitor tools, seemingly high performance suggest datasets are just easy to classify.

Minor comment:

If possible, it will be interesting to interpret the trained model by inspecting attention layers.

Reviewer #2 (Remarks to the Author):

-What are the noteworthy results?

The authors developed a deep learning-based model called MAVERICK to predict the pathogenic variants on Mendelian diseases. They used contextual protein sequence, conservation information and previous annotation of a variant to predict its inheritance trait specific pathogenicity and demonstrated that their model performed better than other pathogenic variant prioritization tools. Another interesting result is that they found that the annotations of the variants from multiple sources which they called structure information was indispensable, while the contribution of variant's sequences is insignificant. Both in simulated and real patients with Mendelian disease, the authors claimed that their model could only use genotype information to prioritize the causal variants on Mendelian diseases. Finally, their model could be generalized to novel Mendelian disease genes.

-Will the work be of significance to the field and related fields? How does it compare to the established literature? If the work is not original, please provide relevant references.

Variants prioritization is critical to Mendelian disease discovery and diagnosis, distinguishing the disease-causing variants from the candidate variants for Mendelian disease is challenging. As the authors showed in the reference list, various tools for prioritizing different types of variants in Mendelian disease have been developed. What is interesting about this manuscript is that compared to existing

tools, MAVERICK could discriminate between variants with different inheritance traits. In addition, they could evaluate more types of protein-altering variants. However, some parts of the methodology and analysis may be problematic.

-Does the work support the conclusions and claims, or is additional evidence needed?

1. In line 26, the authors claimed that MAVERICK could rank the pathogenic variants within the top five variants in over 95% of cases. Then, they claimed 76% of cases were resolved outright. The authors should clarify what does “resolved outright” mean, maybe change it to “on the first guess” as shown in line 252.

2. In lines 140-141, the authors claimed that MAVERICK performs best on loss-of-function variants such as stop-gains and frameshifts, however, as shown in figure 2D, missense SNVs have higher average MAVERICK score than stop-gains, and nonframeshifts have higher average MAVERICK score than frameshifts.

3. In line 160 and other parts of the manuscripts, the authors used known genes/variants to describe genes/variants in the training dataset, and used novel genes to describe genes in the test dataset. I think this might confuse the readers, to my understanding, the known genes/variants are all of the genes/variants in the public databases, such as ClinVar, while the novel genes/variants are those predicted by MAVERICK which are not in the public databases.

4. In line 159-160, the authors claimed that “MAVERICK correctly expects missense variants on SPAST to cause dominant disease even when they are distant from any known disease-causing variant in ClinVar.”, the authors should provide more evidence to support that the predicted dominant disease-causing variants are correctly identified rather than false positives. In addition, In figure 2E, how many high scoring variants not in ClinVar are false positives?

5. In lines 190-191, the authors claimed that PrimateAI overclassifies benign variants as pathogenic because PrimateAI understands the meaning of pathogenicity, to my understanding, if PrimateAI understands the meaning of pathogenicity, it won't overclassify the benign variants as pathogenic ones. The authors should demonstrate that whether they are using PrimateAI correctly.

6. In line 172, MAPPIN predicted >90% of benign variants to be pathogenic, in other words, MAPPIN largely could not distinguish pathogenic variants from benign. However, in the MAPPIN paper, the

MAPPIN could predict correctly for 70% nonsynonymous single nucleotide variants. The authors are suggested to double-check the correctness of the MAPPIN model used in the manuscript and explain why MAPPIN performs so poorly.

-Are there any flaws in the data analysis, interpretation and conclusions? Do these prohibit publication or require revision?

1. In lines 473-475, in the testing dataset, the 16,012 variants fall into the genes which already exist in the training dataset, and as shown in the supplementary figure 3D, structure information (including various gene annotation scores) is the most important contributor to the model's performance, so there may be extensive overlapping genes information between the training and testing datasets, thus it is not surprising that the testing accuracy is high.

-Is the methodology sound? Does the work meet the expected standards in your field?

1. In line 98, the statement "MAVERICK has primarily been evaluated using three datasets" maybe inaccurate. In practice, the validation dataset is used to select the hyperparameters of the model and the test dataset is used to evaluate the performance of the trained model. In addition, in Figure 2 A-B, the validation bars are not necessary.

2. In lines 98-100, the training and validation dataset is split into an $\sim 100:1$ ratio which may be too high, thus the model could be overfitting.

3. In line supplementary figure 1, the output layer is softmax function to model multiclass classification, however, in line 538, the authors used binary cross-entropy as the loss function, in practice, the cross-entropy loss rather than the binary cross-entropy loss should be used.

-Is there enough detail provided in the methods for the work to be reproduced?

1. In Figure 1A, the model structure is misleading, it looks like a multiple layers perception model rather than an ensembled transformer model.

2. In line 125, why the random guessing threshold is 0.1?

3. In lines 396 and 468, The author downloaded the ClinVar reports for January 2020 and 2021, however, there are 4 and 5 different versions of each report, please provide the date for each ClinVar dataset.

4. In line 464, in the validation step, the authors should provide the details of hyperparameters, for example, what hyperparameters need to be validated? How do the models with different hyperparameters perform?

5. In the methods section, the authors are suggested to demonstrate how to calculate precision, recall and F1 score.

Reviewer #3 (Remarks to the Author):

In their manuscript "Deep structured learning realizes variant prioritization for Mendelian diseases" Danzi et al. describe a classification (dominant pathogenic, recessive pathogenic, benign) method for genetic variants in coding sequence (missense, frameshift, nonsense). The method is based on an ensemble of 8 neural network models. The models differ in their number of parameters and to some extent in the features being used. The models are being trained on pathogenic ClinVar variants and a combination of ClinVar benign variants plus variants observed in homozygous state in at least two gnomAD individuals. The authors present several validation and benchmarking results and highlight that their model has been used for prioritizing the disease causal variants in a number of Mendelian samples already. They also outline how their method can be used in combination with prior information about the Mendelian disease type (dominant/recessive) and information about the disease phenotype in terms of associated HPO terms. The manuscript is clearly an interesting read, but in terms of methods not really a game changer in the field. If it was to be published in Nature Communications it would be a very visible publication of this work. In this case, I believe it would require some major improvements in terms of understanding the benefits of an ensemble approach and it would be important to substantially improve on the breadth of benchmarks with other tools.

Here a number of major comments about the work:

- The authors highlight their method as particularly novel (as everyone does) and mention a very limited number of tools that they can compare too. The limited options of 1:1-comparisons are mostly caused by splitting pathogenic variants into dominant and recessive ones in this work (under the "Mendelian" label). I personally have some concerns about such dominant versus recessive differentiation, and I am not sure that it will stand the future of research. We already observe subtle phenotypes in single haplotype carriers of recessive variants, if the patients are phenotyped deeply. Also the whole idea of incomplete penetrance and reduced expressivity for dominant variants will need to grow into a uniform understanding of disease as a continuum of expression effects in relevant pathways – from what we currently distinguish as Mendelian to complex diseases. However, this is beyond the point of this review. I understand that the distinction between recessive and dominant can help, but I personally prefer a continuous scoring of variants rather than classification. On continuous scores, recessive variants naturally score between benign and dominant variants. In this context also a much wider variety of pathogenicity predictors exist (including for non-coding sequence effects and structural variation, which are not covered by this "new score"). I think it is drawing the net too narrow if no comparison is done to tools that do distinguish for example two rather than three classes. Along those lines, this approach is by no means the first one to apply deep learning in pathogenicity prediction. Already in 2015, a group for example proposed an alternative model called DANN on the CADD training dataset. CADD is also an example that InDel and multi-allelic substitution changes (including for frame shifts and nonsense variants) could already be scored by tools in 2013.

- I was a little confused what the authors are trying to imply (about another model based on deep learning), when they say "PrimateAI, which is a more general non-Mendelian pathogenicity classifier", "PrimateAI was developed in a complex inheritance model" or "PrimateAI struggles by overclassifying the benign variants as pathogenic. This is reasonable for PrimateAI and could even be accurate in *the context in which PrimateAI understands the meaning of pathogenicity.*" I think this is somewhat about the point that I am trying to make in the previous section, and I believe it needs to be addressed differently than done by the authors right now.

- It is very interesting to read that their models seem to make good use of population variation derived measures of sequence constraint, while the surrounding primary sequence context seemed of lesser importance to the classification performance. What confused me though is that the models contain the reference and the alternative sequence context (which by definition are usually very similar in a wide 200 amino acid context) and I am not sure that the authors trained models that actually excluded both for the same model. In this context, I am also wondering how reference and alternative sequences are matched in their length / "alignment" towards each other for insertion/deletion changes?

- The authors make the claim that the ensemble approach adds stability and predictive power to their approach. They should quantify this. I think it is relevant to assess the individual models and provide some comparison of how the two different architectures perform as well as the whether the other modifications to the model input make a relevant difference. The application of their model would be considerably cheaper if not all models were required for calculation. The parameters chosen for these

architectures or even just the number of models combined here seems very adhoc. Providing an analysis of these different models might give some indication of whether the authors intuitions were onto something that might generalize for other approaches.

- The manuscript highlights the importance of using variants in new disease genes for model validation. I generally agree. There are limitations to the "historic" ascertainment of such variants though, as genes reported at a specific point in time might also differ in their functional annotations or simply in their level of sequence conservation, to stay with an example. It is very plausible that "older" genes reported in OMIM (or ClinVar) are the ones with strong evolutionary conservation – allowing for their identification before the many advances to the human reference genome and for their study in model systems. Rather than using recently added variants/genes, it might therefore be advantageous to sample from all disease genes, but to use individual genes only in training or validation. In this context it is also important to point out that variants in the same gene, especially those in close proximity, share many of their annotations and most likely also the predicted class. This also brings up another point: it might make sense to match the number of variants in each gene/region between the pathogenic and the benign set as well as to reduce overrepresentation of certain genes in the training/validation sets.

- The presented approach is based on the GRCh37 genome build (btw. the authors use hg19 in some sections of the methods, either use hg19/GRCh37 or always use GRCh37). After almost 10 years of GRCh38, it seems rather unreasonable to have no bona fide implementation of a newly published approach for the current reference genome. I believe it is insufficient to provide a liftOver file of some prescored variants. People have to be able to also score indel variation for the current genome build.

Some minor comments:

- Concluding "a relatively broad representation of ancestries" from the countries that submit to ClinVar is problematic. Even within countries access to genetic testing might be limited to very specific ethnic groups. I do not believe this statement is relevant for the manuscript and I would recommend removing it.

- The authors use "referent" sequence instead of "reference" at several occasions, please correct to "reference".

- For insertions, do you use the maximum score of the two neighboring positions as feature values? Please specify in the methods.

- The authors have some problem with the ALoFT comparison, eventually limiting it to SNVs. I believe the authors should reach out to the original authors and ask for help in obtaining the missing scores.

- Definition of training labels (especially the definition of benign variants) is only clear from the methods. Try to bring this up earlier.

- I really like the analysis on incorporating phenotype information using HPO terms. This is a straightforward approach which could be done with many different methods for benchmarking. I believe it could be described a little better in the main text though – the figure is clear and it would be good if a reader can follow without checking the figure. Further, the difference between applied methods might not be known to most readers (i.e., GADO vs Phenix). It would be good to summarize that.

- Why are variants in the GENESIS cohort removed at an AF of 0.5% while 1% is used for gnomAD? It is the smaller database with the larger uncertainty of the AF estimate. There is also the question whether its disease representation is sufficiently diverse to exclude the filtering of disease causal variants at this level.

POINT BY POINT ANSWER TO REVIEWER COMMENTS

We thank the editor and reviewers for their efforts in reviewing our manuscript. We have been able to respond to the comments in great detail and provide extensive additional data, figures, tables, and code. We hope this revised version of our manuscript will satisfy the questions raised. All changes in the manuscript are redlined.

Reviewer #1 (Remarks to the Author):

In this manuscript, the authors present MAVERICK, a new method for variant prioritization in Mendelian disease. The method uses deep learning based on transformer architectures incorporating various information such as amino acid sequences, allele frequencies, gene constraints etc. As a general comment, I think the methodology part is sound, using a reasonably constructed network architecture. However, I have concerns on the evaluation part, especially on the comparison with Exomiser, a de facto standard tool for the same task.

Comment 1:

The comparison with Exomiser uses the dataset of only 18 patients obtained by the authors' group.

I have a concern that these patients may be cherry picked for MAVERICK to outperform Exomiser.

The author should additionally perform a benchmark using a larger dataset broadly collected from public databases, as currently done for other tools like MAPPIN, ALoFT, and PrimateAI. While the authors avoid this by saying "obviously oversimplifies issues", I am not so convinced.

We agree with the reviewer that the comparison with Exomiser in Figure 5E was far from comprehensive. We generally do not see MAVERICK as a direct competitor to Exomiser due to MAVERICK being a genotype-only tool while Exomiser largely focuses on phenotype, which is why this comparison was so small in scale. But in response to this request, we have now expanded the comparison with Exomiser to 528 patients (a subset of the 644 used in Figure 5A-C) in addition to the 18-patient comparison which is still presented in Figure 5E. The HPO terms used as input for Figure 5E come directly from patient medical notes rather than being general terms chosen based on the causative gene (as done in Figure 5A-C and this new comparison with Exomiser). We believe these patients represent a challenging real-world scenario that is still useful to present as a use case, which we would like to keep in the manuscript. For the analysis with 528 patients, we used a subset of the 644 patients from Figure 5A-C in order to make the comparison as direct and fair as possible:

- 125 patients with causative variants in Maverick's training set where the causal variant is also in Exomiser's whitelist
- 287 patients with causative novel variants in genes known to Maverick, where the causal variant is not in Exomiser's whitelist, but the gene-phenotype relationship still exists in Phenix

- 116 patients with causative variants in genes novel to Maverick, where the causal variant is not in Exomiser's whitelist either and the gene does not necessarily have a known gene-phenotype relationship in Phenix.

This comparison showed that MAVERICK and Exomiser perform essentially equivalently at identifying causal variants on known disease genes (regardless of whether they are in the training set / whitelist or not), while MAVERICK outperforms Exomiser in the identification of causal variants from novel disease genes. This result is in agreement with the result from the smaller analysis of 18 patients in Figure 5E, though the magnitude of difference between the tools is much smaller here. This is not surprising since the cases shown in Figure 5E were selected due to their extremely challenging nature. This can also be seen by how much worse MAVERICK performs in Figure 5E than in Figure 5C.

We have also performed a larger benchmark of MAVERICK against commonly used pathogenicity classifiers (mostly drawn from dbNSFP, but also incorporating MAPPIN). This benchmark was done by extending the spike-in analysis performed in Figure 4B,E to these other tools. This analysis shows each tool's ability to prioritize the causal genetic variant from an individual. The comparisons were performed for genotype-only information, with known inheritance, with incorporation of phenotype information (using GADO), and with incorporation of both inheritance and phenotype information (analogous to Figure 4E). MAVERICK outperformed all other tools in this task in each of these scenarios, but the gap was unsurprisingly the largest in the genotype-only setting.

Comment 2:

The problem setting of MAVERICK is strict. It focuses on Mendelian disease rather than polygenic disease, uses genotype information rather than phenotype information (though optional), and predicts the mode of inheritance rather than benign/pathogenic classification. This strictness makes MAVERICK only comparable with few tools, eliminating many conventional tools for variant prioritization. Thus, the authors should explain more why this problem setting is useful. For example, is it reasonable to assume Mendelian disease and phenotype information is unavailable etc?

With MAVERICK, we thought to develop a tool that can help to close the diagnostic gap for genes with strong genetic effect sizes, i.e., high penetrance. These are (typically) the Mendelian genes as listed in resources such as OMIM and tested in clinical laboratories. For such genes the question of recessive or dominant is essential for the understanding of pathomechanism. For instance, a high constraint pLI score is important for dominant alleles but nearly negligible for recessive loci. We thus gave MAVERICK the ability to score the dominant versus recessive trait as part of its evaluation of pathogenicity. A high score in either category is indicative of higher pathogenicity prediction. Further, there is a line of thought called "genotype first", where medical providers may see comprehensive genetic reports prior to phenotypic assessment; in some regard this is reflected in the discussion around "incidental genetic findings" in exomes and genomes, where phenotypic information was not known (eg undetected cancer). We are not proposing MAVERICK to be used in these instances, but merely pointing out potential use cases for such an approach.

We acknowledge that MAVERICK has limited use for loci with small genetic effects – it is simply not trained for the spectrum of odds ratios seen in GWAS studies for instance. We have, however, attempted to provide a more thorough comparison with other conventional tools for variant prioritization in this revised manuscript in order to demonstrate the utility of MAVERICK.

Comment 3:

The authors should avoid expressions like "excellent performance" when describing benchmark results without competitor tools (e.g. Fig 2). In my opinion, what we can only say in the evaluation of a computational tool is "better or worse" compared to other tools, not absolute "good or bad".

Without competitor tools, seemingly high performance suggest datasets are just easy to classify.

We agree and adjusted word choices as redlined in the revised manuscript.

Minor comment:

If possible, it will be interesting to interpret the trained model by inspecting attention layers.

We thank the reviewer for this thoughtful suggestion, but at this point we believe this is beyond the scope of this article.

Reviewer #2 (Remarks to the Author):

-What are the noteworthy results?

The authors developed a deep learning-based model called MAVERICK to predict the pathogenic variants on Mendelian diseases. They used contextual protein sequence, conservation information and previous annotation of a variant to predict its inheritance trait specific pathogenicity and demonstrated that their model performed better than other pathogenic variant prioritization tools. Another interesting result is that they found that the annotations of the variants from multiple sources which they called structure information was indispensable, while the contribution of variant's sequences is insignificant. Both in simulated and real patients with Mendelian disease, the authors claimed that their model could only use genotype information to prioritize the causal variants on Mendelian diseases. Finally, their model could be generalized to novel Mendelian disease genes.

-Will the work be of significance to the field and related fields? How does it compare to the established literature? If the work is not original, please provide relevant references.

Variants prioritization is critical to Mendelian disease discovery and diagnosis, distinguishing the disease-causing variants from the candidate variants for Mendelian disease is challenging. As the authors showed in the reference list, various tools for prioritizing different types of variants in Mendelian disease have been developed. What is interesting about this manuscript is that compared to existing tools, MAVERICK could discriminate between variants with different inheritance traits. In addition, they could evaluate more types of protein-altering variants. However, some parts of the methodology and analysis may be problematic.

We agree with this assessment, especially in regard to the ability of MAVERICK to assess distinct inheritance traits.

-Does the work support the conclusions and claims, or is additional evidence needed?

1. In line 26, the authors claimed that MAVERICK could rank the pathogenic variants within the top five variants in over 95% of cases. Then, they claimed 76% of cases were resolved outright. The authors should clarify what does "resolved outright" mean, maybe change it to "on the first guess" as shown in line 252.

We agree that this was unclear wording and have changed it to 'solved by the top-ranked variant'.

2. In lines 140-141, the authors claimed that MAVERICK performs best on loss-of-function variants such as stop-gains and frameshifts, however, as shown in figure 2D, missense SNVs have higher average MAVERICK score than stop-gains, and nonframeshifts have higher average MAVERICK score than frameshifts.

We thank the reviewer for this insightful comment. Due to the lack of consistent superior performance by MAVERICK for loss-of-function variants across both the known and novel genes test sets, we have revised this statement to temper our conclusions. It now states that “MAVERICK performs acceptably across all tested variant types”.

3. In line 160 and other parts of the manuscripts, the authors used known genes/variants to describe genes/variants in the training dataset, and used novel genes to describe genes in the test dataset. I think this might confuse the readers, to my understanding, the known genes/variants are all of the genes/variants in the public databases, such as ClinVar, while the novel genes/variants are those predicted by MAVERICK which are not in the public databases.

We regret the confusion around this terminology and have re-written the section introducing these data sets in order to be more clear. The reviewer’s interpretation is not correct though. The training set, the validation set, the known genes test set, and the novel genes test set all came from ClinVar.

4. In line 159-160, the authors claimed that “MAVERICK correctly expects missense variants on SPAST to cause dominant disease even when they are distant from any known disease-causing variant in ClinVar.”, the authors should provide more evidence to support that the predicted dominant disease-causing variants are correctly identified rather than false positives. In addition, In figure 2E, how many high scoring variants not in ClinVar are false positives?

This is a valuable comment. At this moment, we are not able to provide detailed genetic (from affected families) or mechanistic (from experiments) evidence for the actual pathogenicity of the MAVERICK predictions on SPAST. However, we thought it is of great interest to see that 1) the Maverick predictions highly correlate with the known dominant pathogenic changes as well as the non-pathogenicity of known benign changes. Specifically, SPAST has 61 pathogenic missense variants in ClinVar. Of those, 21 are present in MAVERICK’s training set. Maverick correctly classifies all 21 of those training variants as dominant pathogenic. It also classifies 37 of the 40 novel pathogenic variants on SPAST correctly. Additionally, SPAST has 10 benign missense variants in ClinVar. Of those, 6 were in MAVERICK’s training set. Maverick correctly classifies all 10 variants as benign. The figure thus extrapolates this observation to the unknown/ unobserved allelic space of the SPAST gene. Mechanistic or genetic studies in the future, will be able to obtain guidance from these MAVERICK scores.

5. In lines 190-191, the authors claimed that PrimateAI overclassifys benign variants as pathogenic because PrimateAI understands the meaning of pathogenicity, to my understanding, if PrimateAI understands the meaning of pathogenicity, it won’t overclassify the benign variants as pathogenic ones. The authors should demonstrate that whether they are using PrimateAI correctly.

We regret that this comment caused confusion. Since both Reviewers 2 and 3 took objection with this characterization of PrimateAI, we have revised that section in the manuscript greatly.

First, we are confident that we are using PrimateAI correctly. Scores for PrimateAI were downloaded from the PrimateAI Github page (version 0.2), so we did not need to actually run the PrimateAI program in order to produce the results shown.

Second, PrimateAI was developed using a training set that pitted common variants (presumed to be free of selective pressure) in one class against unobserved variants (likely selected against) in the other class. The authors of PrimateAI show that this distinction is strongly correlated with the pathogenic / benign distinction while being free of the biases present in databases like ClinVar. The benign variants in the known and novel genes test sets are ClinVar 1-4 star 'benign' and 'likely benign' variants. They are overwhelmingly rare but are being described in ClinVar as benign primarily in relation to their ability to cause fully penetrant diseases. Our comment about PrimateAI overclassifying these variants as pathogenic was meant as a concession that many ClinVar benign variants have not been evaluated for potential small, low penetrance effects that may cause the kind of negative selection against them that PrimateAI was truly designed to predict.

Due to the confusion that this section caused, we have decided to remove it from the revised manuscript. It has been replaced with a comparison of MAVERICK against dozens of general-purpose pathogenicity classifiers in the variant prioritization task (including PrimateAI). This task assumes that a high penetrance, high effect variant should have a higher score and therefore be ranked more advantageously by a skilled classification system. We believe that this assumption is more valid for PrimateAI and other general-purpose pathogenicity classification tools than our original ClinVar benign vs pathogenic classification task. This variant prioritization task also discards the use of the ClinVar benign variants as a test class for PrimateAI (or any of the other non-Mendelian tools). Furthermore, this task is analogous to tests performed in the PrimateAI paper on samples from the Deciphering Developmental Disorders study.

6. In line 172, MAPPIN predicted >90% of benign variants to be pathogenic, in other words, MAPPIN largely could not distinguish pathogenic variants from benign. However, in the MAPPIN paper, the MAPPIN could predict correctly for 70% nonsynonymous single nucleotide variants. The authors are suggested to double-check the correctness of the MAPPIN model used in the manuscript and explain why MAPPIN performs so poorly.

Pre-calculated MAPPIN scores for every coding position were downloaded from their website, so there should not have been any errors in how the model was run. The MAPPIN paper demonstrates its ability to resolve dominant from recessive pathogenic variants with 70.3% accuracy on the Centers for Mendelian Genomics dataset (64 variants). We were able to replicate this result perfectly using the downloaded pre-calculated MAPPIN scores. Additionally, the MAPPIN paper reports predicting correct inheritance among 78.5% of pathogenic mutations in the Deciphering Developmental Disorders Study dataset (158 variants) as well as predicting overall pathogenicity correctly in 87.3% of them. We were able to replicate these numbers as well using the pre-calculated MAPPIN values. As a result of these replications, we can confidently assert that we did utilize MAPPIN correctly in the manuscript. The discordance between the results we reported, and the results reported in the MAPPIN manuscript is due to the presence of benign variants in the testing set. MAPPIN's ability to discriminate dominant from recessive pathogenic variants is still excellent in the known genes and novel genes test

sets reported in this manuscript (see the recall values in Supplementary Table 2) and is quite close to MAVERICK's performance in that measure.

Are there any flaws in the data analysis, interpretation and conclusions? Do these prohibit publication or require revision?

1. In lines 473-475, in the testing dataset, the 16,012 variants fall into the genes which already exist in the training dataset, and as shown in the supplementary figure 3D, structure information (including various gene annotation scores) is the most important contributor to the model's performance, so there may be extensive overlapping genes information between the training and testing datasets, thus it is not surprising that the testing accuracy is high.

We wholeheartedly agree with this comment. We expect (and see) higher performance on this 'known genes' test set because Maverick got to learn so much about these genes from the training set. But that is why we also do every evaluation with the 'novel genes' test set as well to show that step down in performance explicitly. There is still utility to finding new variants on known genes, which is why we believe the known genes test set is still worth including.

-Is the methodology sound? Does the work meet the expected standards in your field?

1. In line 98, the statement "MAVERICK has primarily been evaluated using three datasets" maybe inaccurate. In practice, the validation dataset is used to select the hyperparameters of the model and the test dataset is used to evaluate the performance of the trained model. In addition, in Figure 2 A-B, the validation bars are not necessary.

We thank the reviewer for this comment and agree that they are correct. The validation dataset performance has now been removed from Figure 2A-B and evaluation has been rephrased to have been done with only the two test sets.

2. In lines 98-100, the training and validation dataset is split into an ~ 100:1 ratio which may be too high, thus the model could be overfitting.

We thank the reviewer for this thoughtful comment. When creating the training dataset, we saw a large effect of training set size on performance. So we chose the 1000 variant validation set to try to maintain as large of a training set as possible while still keeping enough variants held back to accurately perform hyperparameter tuning. We employed other means to mitigate overfitting, such as the use of dropout layers in the model and the one-cycle policy training methodology. Despite this, it may be true that overfitting did still occur. However, this overfitting would not lead to an overestimation of performance on the known and novel genes testing datasets, which maintains the integrity of the presented results.

3. In line supplementary figure 1, the output layer is softmax function to model multiclass classification, however, in line 538, the authors used binary cross-entropy as the loss function, in practice, the cross-entropy loss rather than the binary cross-entropy loss

should be used.

Thank you for catching this typographical error. Line 538 should read 'categorical cross-entropy' as you are correct, this is a three-state classification problem. Categorical cross-entropy is indeed the loss function that was used.

-Is there enough detail provided in the methods for the work to be reproduced?

1. In Figure 1A, the model structure is misleading, it looks like a multiple layers perception model rather than an ensembled transformer model.

We thank the reviewer for this suggestion and have changed the figure into a more accurate depiction of MAVERICK's architecture – an ensemble of transformer models.

2. In line 125, why the random guessing threshold is 0.1?

The random guessing threshold for auPRC on this dataset is 0.1 because the minority class accounts for approximately 10% of the data. This has been clarified in the manuscript as well.

3. In lines 396 and 468, The author downloaded the ClinVar reports for January 2020 and 2021, however, there are 4 and 5 different versions of each report, please provide the date for each ClinVar dataset.

We downloaded the 'variant_summary' report from ClinVar at each timepoint. More detail has been added to the methods section to clarify this.

4. In line 464, in the validation step, the authors should provide the details of hyperparameters, for example, what hyperparameters need to be validated? How do the models with different hyperparameters perform?

We have now provided a new supplementary table (Supplementary Table 7) which describes the hyperparameters used in the two different model architectures. The primary hyperparameters which were tuned were the sizes of the transformers, the number of transformer layers, and the learning rate. Briefly, the size of each transformer was increased until performance plateaued; the number of transformer layers was increased until performance plateaued; and the learning rate was increased until training became unstable and then was decreased to the maximum point of remaining stable. We hope this additional information will be useful, but we believe it is beyond the scope of this article to describe details of how different models performed. This can be easily tested by others with the code we have provided on the MAVERICK Github page.

5. In the methods section, the authors are suggested to demonstrate how to calculate precision, recall and F1 score.

We thank the reviewer for this suggestion and have added these definitions to the methods section.

Reviewer #3 (Remarks to the Author):

In their manuscript "Deep structured learning realizes variant prioritization for Mendelian diseases" Danzi et al. describe a classification (dominant pathogenic, recessive pathogenic, benign) method for genetic variants in coding sequence (missense, frameshift, nonsense). The method is based on an ensemble of 8 neural network models. The models differ in their number of parameters and to some extent in the features being used. The models are being trained on pathogenic ClinVar variants and a combination of ClinVar benign variants plus variants observed in homozygous state in at least two gnomAD individuals. The authors present several validation and benchmarking results and highlight that their model has been used for prioritizing the disease causal variants in a number of Mendelian samples already. They also outline how their method can be used in combination with prior information about the Mendelian disease type (dominant/recessive) and information about the disease phenotype in terms of associated HPO terms. The manuscript is clearly an interesting read, but in terms of methods not really a game changer in the field. If it was to be published in Nature Communications it would be a very visible publication of this work. In this case, I believe it would require some major improvements in terms of understanding the benefits of an ensemble approach and it would be important to substantially improve on the breadth of benchmarks with other tools.

Here a number of major comments about the work:

The authors highlight their method as particularly novel (as everyone does) and mention a very limited number of tools that they can compare too. The limited options of 1:1-comparisons are mostly caused by splitting pathogenic variants into dominant and recessive ones in this work (under the "Mendelian" label). I personally have some concerns about such dominant versus recessive differentiation, and I am not sure that it will stand the future of research. We already observe subtle phenotypes in single haplotype carriers of recessive variants, if the patients are phenotyped deeply. Also the whole idea of incomplete penetrance and reduced expressivity for dominant variants will need to grow into a uniform understanding of disease as a continuum of expression effects in relevant pathways – from what we currently distinguish as Mendelian to complex diseases. However, this is beyond the point of this review. I understand that the distinction between recessive and dominant can help, but I personally prefer a continuous scoring of variants rather than classification. On continuous scores, recessive variants naturally score between benign and dominant variants. In this context also a much wider variety of pathogenicity predictors exist (including for non-coding sequence effects and structural variation, which are not covered by this "new score"). I think it is drawing the net too narrow if no comparison is done to tools that do distinguish for example two rather than three classes. Along those lines, this approach is by no means the first one to apply deep learning in pathogenicity prediction. Already in 2015, a group for example proposed an alternative model called DANN on the CADD training dataset. CADD is also an example that InDel and multi-allelic substitution changes (including for frame shifts and nonsense variants) could already be scored by tools in 2013.

We thank the reviewer for this insightful discussion. We largely agree with the views they have expressed here. The distinctions among dominant, co-dominant, recessive, and incompletely penetrant variants can often be murky and certainly many mis-annotations exist currently. Perhaps the field will move to a more unified view in time, but for now we think this is a useful practical distinction when one is searching for the Mendelian variant of interest in a patient.

We agree with the reviewer that perhaps we “drew the net too narrow” by not comparing MAVERICK to the full complement of standard pathogenicity prediction tools. We have now performed comprehensive assessments of the ability of 36 binary pathogenicity prediction tools to identify pathogenic Mendelian variants spiked into control individuals (Figure 4E-F, Supplementary Figure 6). We also included MAPPIN in this comparison as well as binary versions of both MAVERICK and MAPPIN to quantify the effect that their dominant vs recessive pathogenic distinction has on their ability to prioritize the causal Mendelian pathogenic variant in an individual. We found that MAVERICK outperformed all other tools. The binary version of MAVERICK was the second-best performing tool, suggesting that the dominant vs recessive distinction is not the only facet of MAVERICK that helps it excel at this ranking task.

I was a little confused what the authors are trying to imply (about another model based on deep learning), when they say "PrimateAI, which is a more general non-Mendelian pathogenicity classifier", "PrimateAI was developed in a complex inheritance model" or "PrimateAI struggles by overclassifying the benign variants as pathogenic. This is reasonable for PrimateAI and could even be accurate in *the context in which PrimateAI understands the meaning of pathogenicity.*" I think this is somewhat about the point that I am trying to make in the previous section, and I believe it needs to be addressed differently than done by the authors right now.

Yes, we believe that the point we were attempting to articulate in these passages is very similar to what this reviewer said in their first comment. Please see also our response to Reviewer 2's comment #5 for more details on our changes to this section and clarification about that passage.

These reviews have caused us to reconsider how we originally evaluated tools such as PrimateAI by including variants that may have small or low-penetrance effects as members of the benign class. To move away from that assertion and toward a setting that more naturally considers the range of values in continuous scoring systems as this reviewer recommended, we have replaced the original PrimateAI classification task with a variant ranking task, which also includes comparison with many other tools as discussed in our response to this reviewer's first comment.

It is very interesting to read that their models seem to make good use of population variation derived measures of sequence constraint, while the surrounding primary sequence context seemed of lesser importance to the classification performance. What confused me though is that the models contain the reference and the alternative sequence context (which by definition are usually very similar in a wide 200 amino acid context) and I am not sure that the authors trained models that actually excluded both for the same model. In this context, I am also wondering how reference and alternative

sequences are matched in their length / "alignment" towards each other for insertion/deletion changes?

The reviewer is correct that we did not ever run Maverick with both the reference and the alternate sequences ablated. We have now extended Supplementary Figure 3 with additional ablations of 1) no reference or alternative sequence, 2) no reference, alternative sequence, or ProtT5 embedding of alternative sequence (i.e., structured data only), and 3) all inputs ablated. These results further reinforce that indeed the structured information is the most important component. In fact, they suggest that the structured data is approximately as important as the other three model inputs combined. We also show the 'no inputs' scenario to demonstrate that this ablation methodology is working properly and to show where random guessing performance lies on these plots for each test set.

As for aligning the reference and alternative sequences, there are three parts to the answer. 1) In the case of deletions, we mark amino acids as deleted but the system can still see what they were. Specifically, there is a row in the input matrix right after the one-hot encoding of amino acid which always has a value of 1 for amino acids that exist. In the case of deletions or stop-gains, this value gets set to 0, but the amino acid sequence and conservation data remain visible. 2) For inserted residues, there is no sequence conservation, which shows that residues were inserted. 3) Steps 1 and 2 can indeed create mis-aligned reference and alternative sequences. Before they are compared to each other, a fixed size representation of the reference and alternative sequences are made based off the values at the center of the sequence, since the sequence is centered on the variant being interrogated. But ultimately, the reference and alternative sequences aren't necessarily perfectly aligned, nor is the sequence processed by ProtT5 always exactly the same as the alternative sequence used by the rest of the model in cases where the variant affects the beginning or end of the amino acid sequence or causes a stop gain. These comparisons are instead facilitated by comparing fixed-size representations. The full algorithm can be seen in the code available in the Github repository.

The authors make the claim that the ensemble approach adds stability and predictive power to their approach. They should quantify this. I think it is relevant to assess the individual models and provide some comparison of how the two different architectures perform as well as the whether the other modifications to the model input make a relevant difference. The application of their model would be considerably cheaper if not all models were required for calculation. The parameters chosen for these architectures or even just the number of models combined here seems very adhoc. Providing an analysis of these different models might give some indication of whether the authors intuitions were onto something that might generalize for other approaches.

We thank the reviewer for this suggestion. We have now added a new figure (Supplementary Figure 5C) showing the performance of each individual sub-model on the variant ranking task shown in Figure 3. Model performance is quantified by the area under the curve where the curve plots the cumulative proportion of samples solved by the top-k guesses (as in Figure 3B). The data are normalized so an area under the curve value of 1 would correspond to a model solving every sample on the first guess.

- The manuscript highlights the importance of using variants in new disease genes for model validation. I generally agree. There are limitations to the "historic" ascertainment

of such variants though, as genes reported at a specific point in time might also differ in their functional annotations or simply in their level of sequence conservation, to stay with an example. It is very plausible that "older" genes reported in OMIM (or ClinVar) are the ones with strong evolutionary conservation – allowing for their identification before the many advances to the human reference genome and for their study in model systems. Rather than using recently added variants/genes, it might therefore be advantageous to sample from all disease genes, but to use individual genes only in training or validation. In this context it is also important to point out that variants in the same gene, especially those in close proximity, share many of their annotations and most likely also the predicted class. This also brings up another point: it might make sense to match the number of variants in each gene/region between the pathogenic and the benign set as well as to reduce overrepresentation of certain genes in the training/validation sets.

We thank the reviewer for this insightful comment. We agree that the approach they are advocating would work well and may have even been better than the approach we used. Despite this, re-training the model and subsequently re-doing all analyses in the paper or presenting two distinct models in parallel due to this new training paradigm both are beyond the scope of what we believe is currently warranted.

As for “match[ing] the number of variants in each gene/region between the pathogenic and the benign set as well as to reduce the overrepresentation of certain genes in the training/validation sets”, we agree that it is possible such matching could have produced a better model. We endeavored to mimic the Mendelian landscape of pathogenic mutations in our datasets. Some genes harbor many known pathogenic variants, while others have very few. Much of this is likely due to how long ago those genes were discovered and how much they have been studied, etc, but some may be related to true biological differences. We wanted to seize the opportunity to train on both sets as best we could. Additionally, deep learning is so ‘data hungry’ that we would be concerned about the impact of removing so many training examples.

- The presented approach is based on the GRCh37 genome build (btw. the authors use hg19 in some sections of the methods, either use hg19/GRCh37 or always use GRCh37). After almost 10 years of GRCh38, it seems rather unreasonable to have no bona fide implementation of a newly published approach for the current reference genome. I believe it is insufficient to provide a liftOver file of some prescored variants. People have to be able to also score indel variation for the current genome build.

We agree with the reviewer that this was a shortcoming of our tool. We now provide scripts in the Github repository to process GRCh38 VCF files with Maverick so that indels are scorable for that genome version.

Some minor comments:

Concluding "a relatively broad representation of ancestries" from the countries that submit to ClinVar is problematic. Even within countries access to genetic testing might be limited to very specific ethnic groups. I do not believe this statement is relevant for the manuscript and I would recommend removing it.

We have removed this phrase.

The authors use "referent" sequence instead of "reference" at several occasions, please correct to "reference".

We have corrected the spelling.

For insertions, do you use the maximum score of the two neighboring positions as feature values? Please specify in the methods.

Yes, for insertions we use the maximum score of the two neighboring positions as the feature value. For deletions and block substitutions, we use the max of the entire span. This is now mentioned in the methods of the manuscript.

The authors have some problem with the ALoFT comparison, eventually limiting it to SNVs. I believe the authors should reach out to the original authors and ask for help in obtaining the missing scores.

We appreciate this suggestion from the reviewer. We reached out to Dr. Mark Gerstein, the senior author of the ALoFT manuscript. He directed us to Dr. Suganthi Balasubramanian, the primary author of that tool, for questions. Unfortunately, Dr. Balasubramanian never responded to our emails. We made further attempts to get the ALoFT installation to work using other operating systems, but still had no success. We apologize that this continues to be a limitation in our manuscript.

Definition of training labels (especially the definition of benign variants) is only clear from the methods. Try to bring this up earlier.

We have expanded the initial description of the training dataset to describe this earlier in the results section.

I really like the analysis on incorporating phenotype information using HPO terms. This is a straightforward approach which could be done with many different methods for benchmarking. I believe it could be described a little better in the main text though – the figure is clear and it would be good if a reader can follow without checking the figure. Further, the difference between applied methods might not be known to most readers (i.e., GADO vs Phenix). It would be good to summarize that.

We thank the reviewer for this comment and suggestion. We have now expanded the explanation in the results section of that approach and the GADO, Phenix, and HiPhive tools involved.

Why are variants in the GENESIS cohort removed at an AF of 0.5% while 1% is used for gnomAD? It is the smaller database with the larger uncertainty of the AF estimate. There is also the question whether its disease representation is sufficiently diverse to exclude the filtering of disease causal variants at this level.

We thank the reviewer for pointing this out. This had been done to maintain consistency with some queries that had been run on the GENESIS system. But that had no bearing on this manuscript. We have revised that analysis to filter variants at an allele frequency of 1% for variants in GENESIS to be in line with gnomAD. This did not alter any of the results presented in the manuscript.

REVIEWER COMMENTS

Reviewer #1 (Remarks to the Author):

In this revision, the authors have addressed all of my comments.

Reviewer #3 (Remarks to the Author):

Danzi et al. have provided a revision of their manuscript "Deep structured learning realizes variant prioritization for Mendelian diseases". The revision addresses some of the comments of three reviewers, especially a requested more comprehensive comparison to other tools. This includes a comparison with "pathogenicity scores" as well as the extension of another analysis comparing with Exomiser. I really appreciate these analyses. Both evaluations show some advantage of their tool, while also revealing very incremental advantage in other important settings.

Overall, I feel like the authors went for more minimalistic changes to their manuscript. In my last review, I expressed that the manuscript might require a more substantial revision that also substantiates the claimed benefits from the ensemble approach and provided more thorough investigations in their model/parameter choices if it was to be published in a high impact journal. I personally feel that the authors have not done enough in this respect for their revision.

For example, all three reviewers have brought up questions about the exact implementation or pointed out insufficient information to follow the method / to reimplement the approach. While the authors are providing their code base, this does not substitute for a high-quality methods description. As direct example, the authors respond to my question about how the input sequences are aligned towards each other, but do not include any of that information in the manuscript.

Danzi et al. provided scripts to calculate scores for GRCh38 events now, but after a brief look at their Git repository, I do not see that many people will figure out how to actually do that.

The authors have declined the model analysis suggestion of reviewer 2 or to implement important considerations about training/test data composition incl. linked variant information (currently information is bleeding over between variants as well as training and validation sets). At this point, it is an editorial decision whether the manuscript aligns with the expectations of the journal.

POINT BY POINT RESPONSE TO COMMENTS

We appreciate the opportunity to answer remaining questions and address concerns with additional analysis. This document provides a point-by-point response with comments from us in red. The main manuscript and supplementary information document show all changes and additions in red as well. We have now expanded and performed major reanalysis that have substantially improved the quality and reproducibility of the work. The methods section has been expanded by ~950 words to ensure detailed communication of the process. We further have added a new supplementary figure, a supplementary table, and two paragraphs to the results section. The latter **describing the additional new analysis we have now performed** based on the suggestions of reviewers 2 and 3. This analysis **addresses with great effort the potential information leakage between training and test data and also examines the effect of the training/test set split ratio on model performance**. Additionally, we have released an **update to the GitHub repository housing the MAVERICK software aimed at increasing usability of the tool**. While reviewer 2 did not provide any remarks to the author, we believe we have addressed their concerns as relayed in the editor comments. We hope these changes are well-received and would welcome further discussion.

Reviewer #1 (Remarks to the Author):

In this revision, the authors have addressed all of my comments.

We thank this individual for serving as a reviewer and for the helpful feedback they provided in the first round of review. We are pleased that they are satisfied with the changes.

Reviewer #3 (Remarks to the Author):

Danzi et al. have provided a revision of their manuscript "Deep structured learning realizes variant prioritization for Mendelian diseases". The revision addresses some of the comments of three reviewers, especially a requested more comprehensive comparison to other tools. This includes a comparison with "pathogenicity scores" as well as the extension of another analysis comparing with Exomiser. I really appreciate these analyses. Both evaluations show some advantage of their tool, while also revealing very incremental advantage in other important settings.

We thank this reviewer for their diligence in once-again reviewing our manuscript. We are pleased that they appreciated the expanded comparisons in the former submission.

Overall, I feel like the authors went for more minimalistic changes to their manuscript. In my last review, I expressed that the manuscript might require a more substantial revision that also substantiates the claimed benefits from the ensemble approach and provided more thorough investigations in their model/parameter choices if it was to be published in a high impact journal. I personally feel that the authors have not done enough in this respect for their revision.

For example, all three reviewers have brought up questions about the exact implementation or pointed out insufficient information to follow the method / to reimplement the approach. While the authors are providing their code base, this does not substitute for a high-quality methods

description. As direct example, the authors respond to my question about how the input sequences are aligned towards each other, but do not include any of that information in the manuscript.

We would like to point out that our **first revision was quite substantial and amounted to a 14-page point by point answer, far from minimalistic, that appears to have addressed the majority of concerns raised**. Here, we are eager to provide the **additional analyses suggested by this reviewer as they indeed will address concerns of reproducibility and robustness of MAVERICK**. We have **now significantly further expanded the methods section of the manuscript**, including with a discussion of input sequence alignment. We believe that this quite comprehensively explains how data was collected, the model architecture, how the models were trained, and how the analyses were performed.

Danzi et al. provided scripts to calculate scores for GRCh38 events now, but after a brief look at their Git repository, I do not see that many people will figure out how to actually do that.

This is a valuable comment and we have released an update to the GitHub repository aimed at increasing accessibility of the tool. Now, GRCh37 and GRCh38 operations are both performed through a single script and merely require setting a parameter. We also updated the Readme file to give much more detail and instruction and issued an official 1.0 release of the tool in the 'releases' section. Updates were also made to the Google CoLabs and Jupyter Notebooks housed in the repository. We hope these changes allow more people to utilize MAVERICK more easily.

The authors have declined the model analysis suggestion of reviewer 2 or to implement important considerations about training/test data composition incl. linked variant information (currently information is bleeding over between variants as well as training and validation sets). At this point, it is an editorial decision whether the manuscript aligns with the expectations of the journal.

The concern is that since the ablation studies indicated that the structured information was important for MAVERICK's predictions and many of those values will be similar or identical for different variants on the same gene, particularly if the residues are close to each other, that there is then information leakage from the training set into the validation set and known genes test set. We responded in revision #1 that this information leakage was already addressed by the separate 'novel genes' test set, but we respect renewed concerns that the novel genes set was too small to convincingly address the matter.

We have **now gone back and trained a new version of MAVERICK, which we termed CV-MAVERICK**, by taking the original training set and dividing it into five folds (groups) of unique sets of genes, so that no gene has any variants in more than one of the folds. Each of the original MAVERICK ensemble components were then trained again using the same hyperparameters as used before as a five-fold cross-validation over these folds. As a result, each of these models were trained on 80% of the original training data and tested on the held-out 20% of genes. In this way, we are able to make predictions on all genes in the training set as held-out predictions where there is no information leakage between the data used to train the model making the prediction and the genes on which it is being evaluated. **This creates a much larger test set analogous in function to the novel genes test set. Performance on this held-out test set is similar to MAVERICK's performance on the novel genes test set.** This hopefully provides a better

estimate of MAVERICK's performance on future novel disease genes, which do not benefit from any information leakage from the training set. These results are presented in Supplementary Figure 3D and the new Supplementary Table 2.

Testing CV-MAVERICK on the same known genes and novel genes test sets that MAVERICK was tested on (which largely preserve their meaning here, only that each gene in the known genes test set is known to only ~32 of the 40 ensemble components) demonstrates that the hyperparameters used for training MAVERICK are at least reasonably robust: CV-MAVERICK performs only slightly worse than MAVERICK on each of these test sets (0.8% lower mean auPRC on the known genes test set and 2% lower mean auPRC on the novel genes test set). **This close performance is impressive considering that each model comprising CV-MAVERICK was only trained on 80% of the data on which the MAVERICK models were trained.** Together, this suggests that the dramatic training/validation split of approximately 100:1 did not profoundly impact the model performance, the hyperparameters selected were at least not strongly overfit to the validation set, and the results measured by the novel genes test set are a relatively accurate estimate of the performance of MAVERICK in the absence of information leakage from the training set.

We added new paragraphs to the results section, along with a new supplementary figure and table, attempting to address the impact of the training/test data splitting.

We hope this extensive exercise in testing for the impact of potential information leakage will be satisfactory to the reviewers' concerns. If we have misunderstood the concerns of the editor or the reviewers, please elaborate on them and we would be happy to perform additional experiments to address them. Specifically, if the treatment of the hyperparameter choices and effects of ensembling (or anything else) are not satisfactorily addressed, let us know what you would like done. Grid searches over many combinations of hyperparameter values when training the models would be costly, but we are willing to do so if deemed necessary by the reviewers and editor.

REVIEWERS' COMMENTS

Reviewer #3 (Remarks to the Author):

Danzi et al. have provided a second revision of their manuscript "Deep structured learning realizes variant prioritization for Mendelian diseases". The manuscript describes a classification (dominant pathogenic, recessive pathogenic, benign) method for genetic variants in coding sequence. The method is based on an ensemble of 8 neural network models. The models differ in their number of parameters and to some extent in the features being used. The models are being trained on pathogenic ClinVar variants and a combination of ClinVar benign variants plus variants observed in homozygous state in at least two gnomAD individuals. The revision specifically addressed the usability of their software, a more comprehensive documentation of their methods and a cross-validation approach splitting the training data by genes. I would like to acknowledge all these efforts, that have further improved the software and the manuscript.

I think the study is valuable and will spur some discussions. I say that despite some of my criticism of the first manuscript version still holding for this last iteration. I am reiterating these points:

- The manuscript is clearly an interesting read, but in terms of methods also not a game changer. It is by no means the first approach to apply deep learning in pathogenicity prediction, as pointed out earlier. I think it is important to look at some of the strong statements (e.g., in the abstract and title -- "realizes") and to tone them down.
- While I can see that the distinction between recessive and dominant is still deep-seated in the field, I believe it needs to be overcome. Instead of thinking of incomplete penetrance and reduced expressivity as exceptions, the field needs to adapt a uniform understanding of disease as a continuum of expression effects in relevant pathways. Thus, I have personally always preferred a continuous scoring of variants. I think this aspect should be discussed more.
- The authors make the claim that the ensemble approach adds stability and predictive power to their approach. Thus, I believe it is relevant to assess the contribution of individual models in depth. The application of their model would be considerably cheaper if not all models were required for calculation.
- In the latest version, the authors make sure to use individual genes only in training or validation. As pointed out earlier, it could be important to match the number of variants in each gene/region between the pathogenic and the benign set as well as to reduce overrepresentation of certain genes in the training/validation sets. Even after the additional analyses and alterations that the authors have studied, it remains a worry to me whether the model learned ascertainment biases of ClinVar rather than real biology.

REVIEWERS' COMMENTS

Reviewer #3 (Remarks to the Author):

Danzi et al. have provided a second revision of their manuscript "Deep structured learning realizes variant prioritization for Mendelian diseases". The manuscript describes a classification (dominant pathogenic, recessive pathogenic, benign) method for genetic variants in coding sequence. The method is based on an ensemble of 8 neural network models. The models differ in their number of parameters and to some extent in the features being used. The models are being trained on pathogenic ClinVar variants and a combination of ClinVar benign variants plus variants observed in homozygous state in at least two gnomAD individuals. The revision specifically addressed the usability of their software, a more comprehensive documentation of their methods and a cross-validation approach splitting the training data by genes. I would like to acknowledge all these efforts, that have further improved the software and the manuscript.

We thank the reviewer for again taking the time to review our work and offer constructive feedback.

I think the study is valuable and will spur some discussions. I say that despite some of my criticism of the first manuscript version still holding for this last iteration. I am reiterating these points:

- The manuscript is clearly an interesting read, but in terms of methods also not a game changer. It is by no means the first approach to apply deep learning in pathogenicity prediction, as pointed out earlier. I think it is important to look at some of the strong statements (e.g., in the abstract and title -- "realizes") and to tone them down.

We agree with this comment. We never intended to imply that this was the first to apply deep learning in pathogenicity prediction, merely to *Mendelian* pathogenicity prediction. Nevertheless, we have changed the title and toned down such strong statements in the abstract and throughout the manuscript.

- While I can see that the distinction between recessive and dominant is still deep-seated in the field, I believe it needs to be overcome. Instead of thinking of incomplete penetrance and reduced expressivity as exceptions, the field needs to adapt a uniform understanding of disease as a continuum of expression effects in relevant pathways. Thus, I have personally always preferred a continuous scoring of variants. I think this aspect should be discussed more.

We completely respect and appreciate this viewpoint. Our decisions in constructing our scoring scheme were drawn largely out of pragmatism rather than a disagreement with these raised points from a conceptual perspective. We have added several sentences to the discussion acknowledging this limitation.

- The authors make the claim that the ensemble approach adds stability and predictive power to their approach. Thus, I believe it is relevant to assess the contribution of individual models in depth. The application of their model would be considerably cheaper if not all models were required for calculation.

We do agree that in the additional work we have done attempting to examine the performance abilities of the different ensemble members we are seeing more similar performance levels than our earlier tests had suggested. We now offer scripts through the Github to run 'MAVERICK-lite' using only ensemble member 1, which does not use the language model component and is therefore much faster, at a small cost to accuracy. Ensemble member 1 was chosen because it had the highest mean score of area under the precision recall curves for the test sets among the submodels that did not use the language model component, based on data from Figure S5. It is worth noting that the submodels which use the language model component outperformed ensemble member 1, but we do not see the utility in providing them as a 'MAVERICK-lite' option as the compute for the language model is performed only once across all ensemble members and so the marginal cost to run the entire MAVERICK model is small once one model of architecture 2 has been run (approximately 3B parameters vs 3.005B).

- In the latest version, the authors make sure to use individual genes only in training or validation. As pointed out earlier, it could be important to match the number of variants in each gene/region between the pathogenic and the benign set as well as to reduce overrepresentation of certain genes in the training/validation sets. Even after the additional analyses and alterations that the authors have studied, it remains a worry to me whether the model learned ascertainment biases of ClinVar rather than real biology.

We appreciate this concern, but we believe that our use of the novel genes test set as well as the cross-validation tests provide very strong evidence against this hypothesis without diminishing the size of the training set as dramatically as would be done by following this suggestion.